# Modulation of M2 macrophage polarization by the crosstalk between Stat6 and Trim24

Tao Yu [1,8], Shucheng Gan[1,8], Qingchen Zhu[1,8], Dongfang Dai[2], Ni Li[1,3], Hui Wang[4], Xiaosong Chen[4], Dan Hou[1], Yan Wang[1], Qiang Pan[1,3], Jing Xu[1], Xingli Zhang[1], Junli Liu[1], Siyu Pei[1], Chao Peng [5], Ping Wu[5], Simona Romano [6], Chaoming Mao[2], Mingzhu Huang[7], Xiaodong Zhu[7], Kunwei Shen[4], Jun Qin [1,3]* & Yichuan Xiao [1]*

Stat6 is known to drive macrophage M2 polarization. However, how macrophage polarization is fine-tuned by Stat6 is poorly understood. Here, we find that Lys383 of Stat6 is acetylated by the acetyltransferase CREB-binding protein (CBP) during macrophage activation to suppress macrophage M2 polarization. Mechanistically, Trim24, a CBP-associated E3 ligase, promotes Stat6 acetylation by catalyzing CBP ubiquitination at Lys119 to facilitate the recruitment of CBP to Stat6. Loss of Trim24 inhibits Stat6 acetylation and thus promotes M2 polarization in both mouse and human macrophages, potentially compromising antitumor immune responses. By contrast, Stat6 mediates the suppression of *TRIM24* expression in M2 macrophages to contribute to the induction of an immunosuppressive tumor niche. Taken together, our findings establish Stat6 acetylation as an essential negative regulatory mechanism that curtails macrophage M2 polarization.

[1] CAS Key Laboratory of Tissue Microenvironment and Tumor, Institute of Health Sciences, Shanghai Jiao Tong University School of Medicine & Shanghai Institutes for Biological Sciences, Chinese Academy of Sciences, University of Chinese Academy of Sciences, Shanghai 200031, China. [2] Institute of Oncology and Department of Nuclear Medicine, The Affiliated Hospital of Jiangsu University, 438 Jiefang Road, Zhenjiang 212001, China. [3] CAS Center for Excellence in Molecular Cell Science, Shanghai Institutes for Biological Sciences, Chinese Academy of Sciences, Shanghai 200031, China. [4] Comprehensive Breast Health Center, Ruijin Hospital, Shanghai Jiaotong University School of Medicine, Shanghai 200025, China. [5] National Facility for Protein Science in Shanghai, Zhangjiang Lab, Shanghai 201210, China. [6] Department of Molecular Medicine and Medical Biotechnology, University of Naples, Federico II, 5-80131 Naples, Italy. [7] Department of Medical Oncology, Fudan University Shanghai Cancer Center, Shanghai, China. [8] These authors contributed equally: Tao Yu, Shucheng Gan, Qingchen Zhu. *email: qinjun@sibs.ac.cn; ycxiao@sibs.ac.cn

Macrophages, a type of functionally diversified immune cell widely spread throughout the whole body of adult mammals, play critical roles in the regulation of homeostasis, inflammation, and antitumor immunity in a tissue-specific and context-dependent manner[1–3]. These cells are commonly classified into classically activated (also called M1) macrophages, which are induced by the type 1 T helper (Th1) cell signature cytokine interferon-γ (IFN-γ) and/or the toll-like receptor (TLR) ligand, and alternatively activated (also called M2) macrophages that result from the stimulation of Th2 signature cytokine interleukin-4 (IL-4) or IL-13[1–3]. In tumors, infiltrating macrophages, also known as tumor-associated macrophages (TAMs), are usually educated by environmental factors to present a M2 state, exerting an inhibitory effect on the cytotoxic function of tumor-killing immune cells and thus impairing antitumor immunity, which in turn contributes to the immunosuppressive tumor niche[4–6]. Therefore, increased TAM infiltration is correlated with the poor outcome of solid tumors in a variety of animal models and human cancer patients[7–9]. Consequently, targeting TAMs, by inhibiting M2 activation, ablating this population, or inducing a proinflammatory M1 state in TAMs, inhibits the progression of tumor growth and metastasis and has thus been applied in the clinic for the treatment of many solid tumors[10–13].

Macrophage M2 polarization involves tyrosine phosphorylation and activation of a signal transducer and activator of transcription 6 (Stat6), which mediates the transcriptional activation of M2 macrophage-specific genes such as arginase 1 (Arg1), mannose receptor 1 (Mrc1), resistin-like α (Retnla, Fizz1), chitinase-like protein 3 (Chil3, Ym1), and the chemokine genes Ccl17 and Ccl24[14]. The essential positive role of Stat6 in macrophage M2 polarization is reflected by the enhanced expression of M2 genes in Stat6-overexpressing macrophages[15], whereas the ablation of Stat6 abolishes M2 gene expression[16]. Moreover, published studies have suggested that genetic deletion or pharmacological inhibition of Stat6 dramatically suppresses tumor growth and promotes the antitumor immune responses of macrophages[17,18]. However, how macrophage polarization is fine-tuned by Stat6 is poorly understood.

The activation of STAT family proteins is associated with tyrosine phosphorylation at specific sites, which promotes the formation of homodimers or heterotrimers with other transcription factors that then translocate into the nucleus to initiate transcription. More recently, accumulating evidence has suggested that lysine acetylation is an additional modulatory mechanism that controls STAT protein activity[19]. The acetylation of STAT proteins, such as STAT1, STAT2, STAT3, and STAT5, at specific lysine sites has been suggested to regulate their DNA-binding affinity, protein–protein interactions, and dimerization[20–24]. Nevertheless, the acetylation sites in Stat6 have not yet been identified, and the biological consequences of this modification in Stat6 remain elusive.

In the present study, we identify that Stat6 is acetylated at Lys383 by the acetyltransferase CREB-binding protein (CBP) during macrophage M2 polarization. Rather than activating Stat6 activity, the acetylation of Stat6 dramatically suppresses its transcriptional activity and thus inhibits macrophage M2 polarization. Interestingly, Stat6 acetylation is found to be controlled by the CBP-associated E3 ligase Trim24, which directly mediates CBP ubiquitination at Lys119 and facilitates the binding of CBP with Stat6. Consequently, Trim24 deficiency dramatically suppresses Stat6 acetylation and thus promotes M2 polarization and impairs the antitumor immune function of macrophages.

## Results

**Stat6 is acetylated at Lys383**. To test whether Stat6 acetylation plays a role during macrophage M2 polarization, we stimulated murine macrophages with IL-4 and then immunoprecipitated Stat6 to examine its acetylation status. IL-4 stimulation induced the substantial acetylation of lysine residues in Stat6 in macrophages (Fig. 1a). Previous studies suggested that CBP, a well-known protein lysine acetyltransferase, mediates Stat6 acetylation[25,26]. Indeed, CBP overexpression in 293T cells directly mediated the acetylation of Stat6 (Fig. 1b). In addition, CBP knockdown abolished the lysine acetylation of Stat6 in immortalized mouse macrophages (Fig. 1c), which validated the critical role of CBP in mediating Stat6 acetylation.

To study the biological function of Stat6 acetylation, we examined Stat6-controlled luciferase activities in 293T cells pretreated with or without nicotinamide (NAM) and trichostatin A (TSA), which are inhibitors of the SIRT family of deacetylases and histone deacetylases (HDACs). The results indicated that TSA/NAM pretreatment promoted Stat6 acetylation without affecting its tyrosine phosphorylation, and Stat6 transcriptional activity was dramatically suppressed (Fig. 1d). In addition, TSA/NAM pretreatment did not affect Stat6 phosphorylation or nuclear translocation in murine primary macrophages (Fig. 1e). These results suggested that Stat6 acetylation negatively regulates macrophage M2 polarization.

DNA is negatively charged because of its phosphate backbone, and positively charged lysine (K) or arginine (R) amino acids in the DNA-binding domain (DBD) of a transcription factor stabilizes its association with a specific DNA sequence. However, acetylation removes the positive charge of the lysine side chain from the transcription factor and thus inhibits its DNA-binding ability[27,28]. Since the above-mentioned results suggested that Stat6 acetylation abolishes its transcriptional activity without affecting its phosphorylation and nuclear translocation, we speculated that this modification in Stat6 occurs at the DBD and directly impairs its DNA-binding activity. Indeed, chromatin immunoprecipitation (ChIP)–quantitative polymerase chain reaction (QPCR) analysis revealed that TSA/NAM pretreatment dramatically inhibited the DNA-binding affinity of Stat6 in the promoters of M2 genes in murine primary macrophages (Fig. 1f). To identify the acetylation site in Stat6, we carried out the mass spectrometry (MS) analysis and identified several potential acetylation sites in Stat6, including Lys73, Lys374, Lys383, and Lys636 (Fig. 1g, Supplementary Fig. 1). Next, we generated the mutant Stat6 by replacing the above lysine residues and all 11 other lysine residues in the DBD with arginine (Fig. 1h, i). This lysine (K)-to-arginine (R) substitution prevents acetylation but maintains a positive charge, thus mimicking the nonacetylated form of a protein. Interestingly, only the Stat6 K383R mutant, but not the other mutants, was not acetylated by following the overexpression of CBP (Fig. 1i), suggesting that the Stat6 acetylation site is Lys383 in the DBD. In addition, although Lys73 and Lys636, which are not located in the DBD, are conserved lysine acetylation sites in the $G(S)KX_{3–5}P$ sequence in STAT family proteins[22], these residues are not the acetylation site of Stat6 (Fig. 1i). Moreover, the Lys383 site is evolutionarily conserved in Stat6 proteins among different species but is present in only Stat6 and Stat5 among all the STAT proteins (Fig. 1j), suggesting the functional specificity of this lysine in Stat6. Taken together, these results identified and verified Lys383 in Stat6 as the site of CBP-mediated acetylation.

**Stat6 K383 acetylation inhibits its transcriptional activity**. To confirm that Stat6 acetylation occurs at Lys383, we generated a polyclonal antibody against Lys383-acetylated Stat6 (Fig. 2a, b). Indeed, this antibody recognized CBP-induced acetylation in Stat6 but not the K383R mutant Stat6, although there was basal background signal due to a nonspecific reaction (Fig. 2c). In

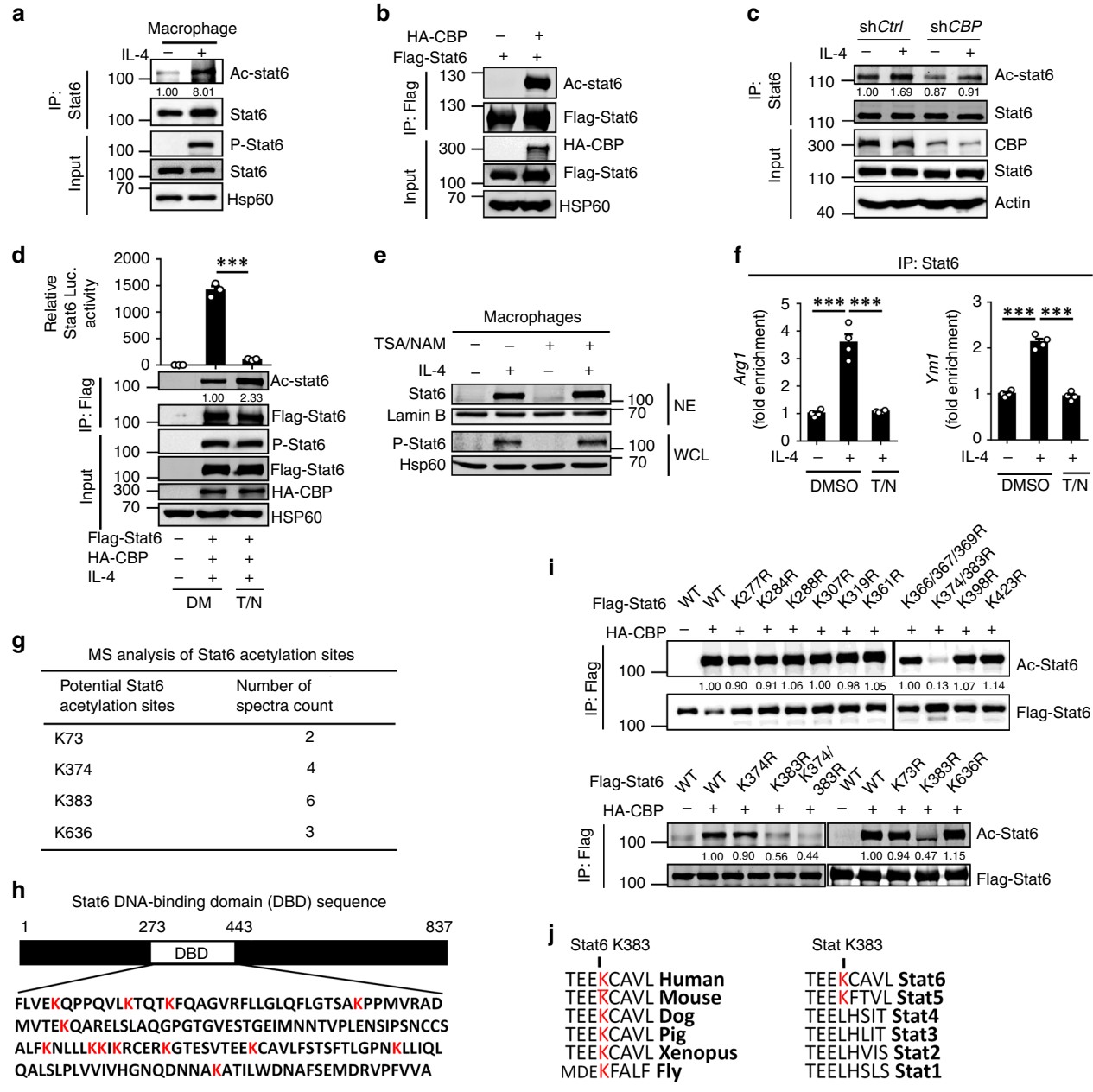

**Fig. 1** Stat6 is acetylated at Lys383. **a–c** Immunoblot analysis of the lysine acetylation of Stat6 in primary murine macrophages (**a**) or 293T cells transfected with the indicated expression vectors (**b**), or in control and *CBP*-knockdown iBMDMs (**c**) that were (+) or were not (−) stimulated with IL-4 for 30 min; lysates were assessed by immunoprecipitation (IP) with anti-Stat6 or anti-Flag and immunoblotting with anti-Ac-Lys and anti-Stat6, or anti-Flag, and immunoblot of the lysates to detect input proteins and loading controls. **d** Luciferase assay showing Stat6 transcriptional activity in 293T cells transfected with the indicated vector that were pretreated with DMSO (DM) or trichostatin A (TSA, 5 μM) plus nicotinamide (NAM, 1 mM) (T/N) and then stimulated with (+) or without (−) IL-4 for 4 h before detection. Lysates were immunoblotted for acetylated (Ac−), phosphorylated (P−) Stat6, HA, Flag, and Hsp60 as controls. **e**, **f** Immunoblot of Stat6 and Lamin B in nuclear extracts and phosphorylated (P)-Stat6 and Hsp60 in whole-cell lysates (**e**) and ChIP–QPCR analysis of Stat6 binding to the promoters of *Arg1* or *Ym1* (**f**) in murine primary macrophages that were pretreated with DMSO or TSA plus NAM (T/N) and then stimulated with (+) or without (−) IL-4 for 1 h. **g** Mass spectrometry analysis showing potential acetylation sites in Stat6 after the immunoprecipitation of Stat6 in 293T cells transfected with Stat6 and CBP. **h** Schematic representation of the mouse Stat6 protein showing the DNA-binding domain (DBD) and its amino acid sequence with all lysine (K) residues highlighted in red. **i** Immunoblot analysis of the lysine acetylation of wild-type (WT) and KR mutant Stat6 in 293T cells transfected with the indicated expression vectors; lysates were assessed by immunoprecipitation (IP) with anti-Flag and immunoblotting with anti-Ac-Lys and anti-Flag. **j** Amino acid sequence alignment of Stat6 among the indicated species and different mouse STAT proteins showing Lys383 that are highlighted in red. Data with error bars are represented as mean ± SD. Each panel is a representative experiment of at least three independent biological replicates. *p < 0.05, **p < 0.01, ***p < 0.001 as determined by the unpaired Student's *t* test. Source data are provided as a Source Data file

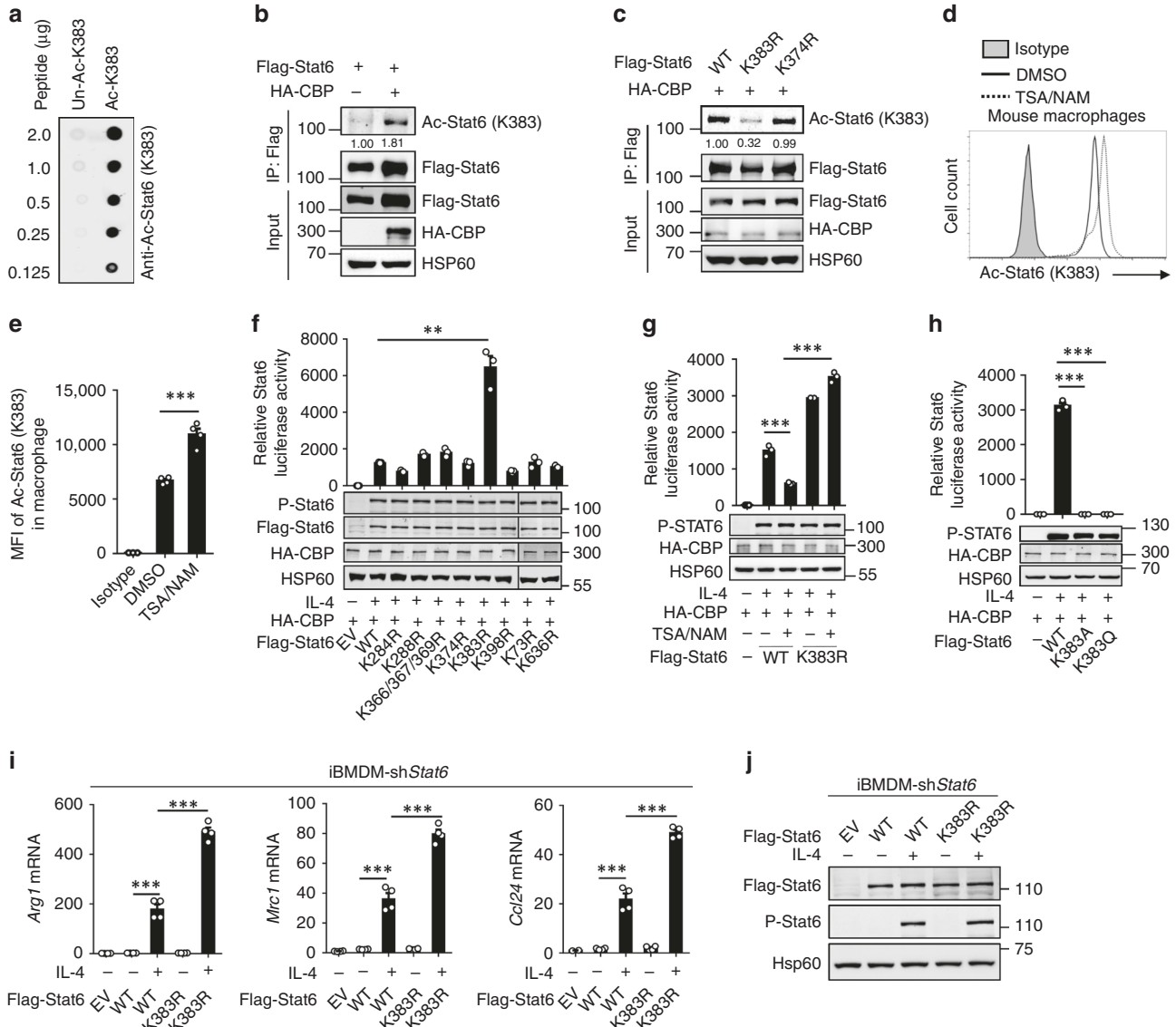

**Fig. 2** Lys383 acetylation in Stat6 suppressed its transcriptional activity. **a** Dot-blot analysis to detect the efficiency of the antibody against Stat6 site-specific (Lys383) acetylation. **b**, **c** Immunoblot analysis of the Lys383 acetylation of Stat6 in 293T cells transfected with the indicated expression vectors; lysates were assessed by immunoprecipitation (IP) with anti-Flag and immunoblotting with anti-Ac-Stat6 (K383) and anti-Flag; lysates were immunoblotted to detect input proteins and loading controls before immunoprecipitation. **d**, **e** Flow cytometric analysis of the endogenous Stat6 acetylation at Lys383 in murine primary peritoneal macrophages isolated from naive C57BL/6 mice treated with DMSO or trichostatin A (TSA) (5 μM) plus nicotinamide (NAM) (1 mM) (T/N) for 2 h. Data are presented as a representative histography (**d**) and summary bar graph (**e**). **f**–**h** Luciferase assay to assess Stat6 transcriptional activity in 293T cells transfected with wild-type (WT) or KR mutant Stat6 (**f**), 293T cells transfected with WT or the K383R mutant Stat6 pretreated with DMSO (−) or with NAM plus TSA (**g**), and 293T cells transfected with WT or K383A or the K383Q mutant Stat6 (**h**) and then stimulated with (+) or without (−) IL-4 (20 ng mL$^{-1}$) for 4 h before detection. Immunoblots showing the protein expression levels in transfected cells are presented below each bar graph. **i**, **j** QPCR analysis of *Arg1*, *Mrc1*, and *Ccl24* mRNA in *Stat6*-knockdown iBMDMs that were reconstituted with empty vector (EV), WT, or the K383R mutant Stat6, and then were left unstimulated (−) or stimulated (+) with IL-4 for 6 h (**i**). Immunoblot of Flag-Stat6, phosphorylated (P)-Stat6, and Hsp60 showing the reconstitution efficiency of Flag-Stat6 (**j**). Data with error bars are represented as mean ± SD. Each panel is a representative experiment of at least three independent biological replicates. *$p < 0.05$, **$p < 0.01$, ***$p < 0.001$ as determined by the unpaired Student's *t* test. Source data are provided as a Source Data file

addition, the deacetylase inhibitor TSA/NAM pretreatment significantly promoted the acetylation of Lys383 in Stat6 in murine primary macrophages (Fig. 2d, e). To study the function of Lys383 acetylation in Stat6, we examined the effect of different Stat6 mutations on the regulation of Stat6 transcriptional activity. As expected, only the presence of the K383R mutant Stat6, but not other DBD KR mutants and the K73R and K636R mutants, dramatically enhanced Stat6-driven luciferase activity (Fig. 2f). In addition, TSA/NAM pretreatment suppressed only wild-type

(termed WT hereafter) Stat6-induced luciferase activity but did not efficiently inhibit K383R mutant-induced luciferase activity (Fig. 2g). We then generated two other Stat6 mutants by replacing Lys383 in Stat6 with glutamine (Q) or alanine (A) and verified their function. K-to-Q or K-to-A substitutions neutralize the positive charge in lysine and thus suppress binding to negatively charged DNA. Accordingly, K383Q or the K383A mutant Stat6 exhibited completely abolished Stat6-driven luciferase activity (Fig. 2h). More interestingly, reconstitution of the K383R mutant

Stat6 in *Stat6*-knockdown macrophages induced the significantly increased expression of M2 genes compared with that in macrophages reconstituted with WT Stat6 upon IL-4 stimulation (Fig. 2i, j), which confirmed the negative role of Stat6 Lys383 acetylation in restraining macrophage M2 polarization.

**Trim24 promotes CBP-mediated Stat6 acetylation**. Since Stat6 acetylation is strongly induced in macrophages upon IL-4 stimulation, we hypothesized that IL-4-regulated genes modulate CBP-mediated Stat6 acetylation. Therefore, we performed RNA-sequencing analysis and identified 1613 up/downregulated genes (Log2 ≤ –0.75 or Log2 ≥ 0.75, $P ≤ 0.05$, false-discovery rate ≤ 0.5) in murine primary macrophages upon IL-4 stimulation. In addition, as CBP-associated proteins may regulate CBP-mediated Stat6 acetylation, we performed MS analysis to identify potential CBP-binding proteins after the immunoprecipitation of CBP. Among the 1613 differentially expressed genes, the Venn diagram indicated that there were 63 genes that encoded CBP-binding proteins as well by mass spectroscopy analysis (unique peptides ≥ 1), suggesting that these genes might be the candidate to regulate CBP-mediated Stat6 acetylation. Among these genes, the most abundantly expressed genes that also have a reported function in macrophages were selected, and Trim24, a bromodomain-containing protein that can recognize acetylated lysine residues, was chosen for further analysis (Fig. 3a). We found that Trim24 was mainly located in the nuclei of macrophages, and IL-4 stimulation did not affect its subcellular localization (Supplementary Fig. 2a). In addition, immunofluorescent images showed that Trim24 was colocalized with CBP in the nuclei of macrophages upon IL-4 stimulation (Fig. 3b). Interestingly, Trim24 alone could not directly induce lysine acetylation in Stat6 but dramatically promoted CBP-induced acetylation in WT, but not the K383R mutant, Stat6 in a dose-dependent manner (Fig. 3c, d).

To study how Trim24 modulates CBP-induced Stat6 acetylation, we examined the association of CBP with Stat6. Interestingly, overexpression of Trim24 promoted the binding of CBP with Stat6 in 293T cells transfected with vectors expressing these proteins (Fig. 3e). We also observed that Trim24 together with CBP and Stat6 formed a trimeric complex in response to IL-4 stimulation, whereas *Trim24* knockdown disrupted the formation of the complex and thus suppressed the IL-4-induced endogenous Stat6 acetylation at Lys383 (Fig. 3f, g). Accordingly, overexpression of *Trim24* dramatically suppressed the transcriptional activity of Stat6, whereas knockdown of *Trim24* significantly enhanced Stat6 transcriptional activity and promoted macrophage M2 polarization (Fig. 3h–j). Taken together, these data established Trim24 as a pivotal positive regulator of CBP-mediated Stat6 acetylation through promoting the association between CBP and Stat6.

**Trim24 mediates CBP ubiquitination**. To investigate how Trim24 regulates the association of CBP with Stat6, the plasmids encoding full-length Trim24 and functional domains-deleted Trim24 were generated (Fig. 4a). Surprisingly, a C-terminal deleted truncated Trim24 construct that lacks the pleckstrin homology domain and the acetylated lysine-recognizing bromodomain exerted an effect similar to that of full-length Trim24 and promoted CBP-mediated Stat6 acetylation. However, an N-terminal deleted truncated Trim24 construct that lacks the Ring domain did not efficiently promote lysine acetylation in Stat6 (Fig. 4b). In addition, deletion of the Ring domain abolished the Trim24-mediated suppression of Stat6 transcriptional activity (Supplementary Fig. 2b).

The Ring domain is essential for the ubiquitination function of E3 ubiquitin ligases[29], so we speculated that Trim24 functions as

an E3 ligase to mediate CBP-induced Stat6 acetylation. Interestingly, IL-4 stimulation did not affect Stat6 ubiquitination (Supplementary Fig. 2c) but dramatically promoted endogenous CBP ubiquitination, and *Trim24* knockdown abolished IL-4-induced endogenous CBP ubiquitination in murine immortalized macrophages (Fig. 4c). In 293T cells, overexpression of full-length Trim24, but not a Ring domain-deleted mutant (Trim24ΔN), markedly enhanced CBP ubiquitination, and Trim24 specifically promoted the Lys63-linked, but not Lys48-linked, ubiquitination of CBP (Fig. 4d, Supplementary Fig. 2d). Consistently, the K63R ubiquitin mutant was unable to mediate Trim24-stimulated ubiquitination of CBP (Supplementary Fig. 2e). Moreover, an in vitro ubiquitination assay revealed that the Trim24 protein could directly add ubiquitin from precharged E2 (UbcH5a) to the CBP protein (Fig. 4e), suggesting that Trim24 is an E3 ligase that directly catalyzes the ubiquitination of CBP.

We next performed MS analysis to identify the ubiquitination site in CBP and found that CBP is ubiquitinated by Trim24 at Lys119 (Fig. 4f). Accordingly, overexpression of Trim24 promoted the ubiquitination of WT, but not the K119R mutant, CBP (Fig. 4g), and an in vitro ubiquitination assay confirmed that the Trim24 protein could not add a ubiquitin chain to the K119R mutant CBP protein (Fig. 4h). Consistently, Trim24 was unable to promote the recruitment of Stat6 to the ubiquitination-free K119R mutant CBP and did not efficiently enhance the mutant CBP-mediated Stat6 acetylation at Lys383 (Fig. 4i, j). However, the K119R mutation in CBP did not affect the association of CBP with Trim24 (Supplementary Fig. 2f), suggesting that Lys63-linked ubiquitination at Lys119 is specifically critical for the binding of CBP with Stat6. These results collectively suggested that Trim24-induced CBP ubiquitination at Lys119 is critical for the recruitment of Stat6 to CBP and thus mediates the acetylation of Stat6.

**Stat6 K383 acetylation restrains macrophage M2 polarization**. To study the in vivo function of Trim24-modulated Stat6 acetylation, we generated conditional knockout (KO) mice in which Trim24 was deleted specifically in myeloid cells (termed $Trim24^{M-/-}$ hereafter) by crossing *Trim24* floxed mice with LysM cre mice (Supplementary Fig. 3). *Trim24* ablation did not affect the development, maturation, and activation of myeloid cells and lymphoid cells in vivo (Supplementary Fig. 4). Consistent with the knockdown data, Trim24 deletion in murine primary macrophages dramatically suppressed IL-4-induced endogenous CBP ubiquitination and Stat6 acetylation at Lys383 (Fig. 5a, b). Furthermore, Stat6 was slightly acetylated in *Trim24*-deficient macrophages upon IL-4 stimulation, implying that other acetyltransferases, such as p300, may function redundantly to mediate Stat6 acetylation in the absence of Trim24-mediated CBP activation. Accordingly, the loss of Trim24 significantly promoted the DNA-binding activity of Stat6, as shown by the electrophoretic mobility shift assay (EMSA) (Fig. 5c). To further examine the binding activity of Stat6 at the genomic level, we immunoprecipitated Stat6 from IL-4-stimulated WT and *Trim24*-deficient murine primary macrophages and analyzed the coimmunoprecipitated DNA with deep sequencing (CHIP-Seq). As expected, the loss of Trim24, which suppressed Stat6 K383 acetylation, enhanced IL-4-induced Stat6 binding to M2 gene promoters (Fig. 5d, e). These data were further validated by ChIP–QPCR results showing that *Trim24* deficiency indeed promoted IL-4-activated Stat6-binding activity in a time-dependent manner (Fig. 5f). In addition, the K383R mutation in Stat6, which abolished Stat6 acetylation, dramatically promoted Stat6 binding to M2 gene promoters compared with that of WT Stat6 in

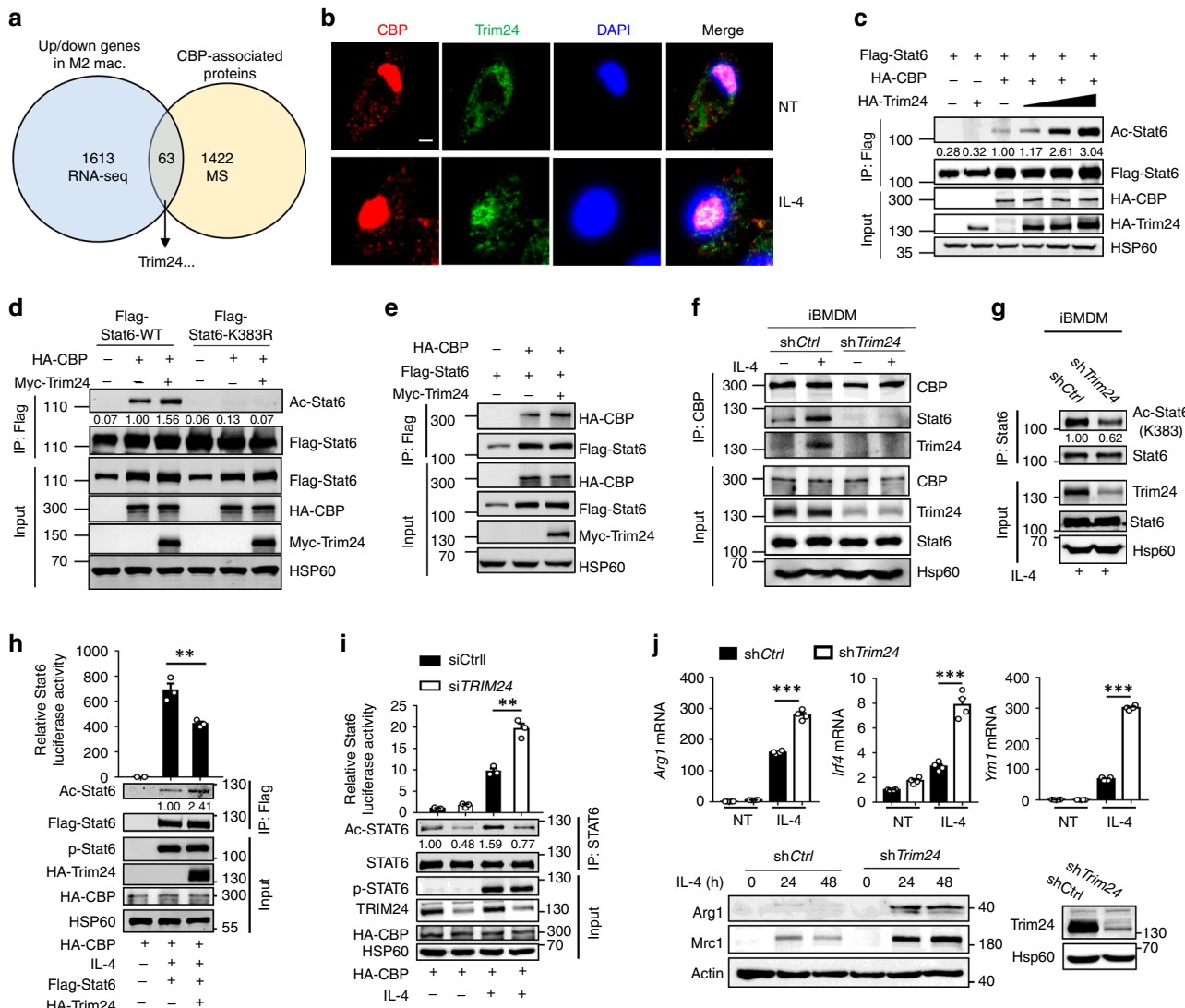

**Fig. 3** Trim24 promoted CBP-induced Stat6 acetylation. **a** Venn diagram showing the number of genes up/downregulated in M2-polarized macrophages and those that encode CBP-associated proteins. **b** Confocal microscopy images showing the intracellular localization of Trim24 and CBP in peritoneal macrophages that were left nontreated (NT) or stimulated with IL-4 for 30 min, with DAPI staining used to indicate the nuclei of the cells. Scale bars: 5 μm. **c**, **d** Immunoblot analysis of the lysine acetylation of Stat6 in 293T cells transfected with the indicated expression vectors assessed by immunoprecipitation (IP) with anti-Flag and immunoblotting with anti-Ac-Lys and anti-Flag. **e**, **f** Interaction between CBP, Stat6, and Trim24 in 293T cells transfected with the indicated expression vectors (**e**) and control and *Trim24*-knockdown iBMDMs that were left unstimulated (−) or stimulated with IL-4 (+) for 30 min (**f**) assessed by immunoprecipitation (IP) with anti-Flag or anti-CBP and immunoblotting with anti-HA and anti-Flag, or anti-CBP, anti-Stat6, and anti-Trim24. **g** Immunoblot analysis of endogenous lysine acetylation of Stat6 in control and *Trim24*-knockdown iBMDMs stimulated with IL-4 for 30 min assessed by immunoprecipitation (IP) with anti-Stat6 and immunoblotting with anti-Ac-Stat6 (K383) and anti-Stat6. **h**, **i** Luciferase assay to assess Stat6 transcriptional activity in 293T cells transfected with the indicated expression vectors (**h**) and 293T cells transfected with scramble (siCtrl) or si*TRIM24* (**i**) stimulated with (+) or without (−) IL-4 (50 ng mL⁻¹) for 4 h before detection. Immunoblot of acetylated (Ac−) Stat6 after immunoprecipitation; phosphorylated (P−) Stat6, HA-CBP, HA-Trim24, or Hsp60 in whole-cell lysates were applied as controls. **j** QPCR and immunoblot analysis of M2 gene expression in control and *Trim24*-knockdown iBMDMs that were left nontreated (NT) or stimulated with IL-4, and the immunoblot of Trim24 and Hsp60 to detect the knockdown efficiency. Data with error bars are represented as mean ± SD. Each panel is a representative experiment of at least three independent biological replicates. *$p < 0.05$, **$p < 0.01$, ***$p < 0.001$ as determined by the unpaired Student's $t$ test. Source data are provided as a Source Data file

immortalized murine macrophages upon IL-4 stimulation (Fig. 5g). These results suggested that Stat6 K383 is critical for Stat6–DNA-binding activity and its binding to M2 gene promoters.

To determine the biological consequences of Trim24-mediated Stat6 K383 acetylation, we performed RNA sequencing to examine Stat6-induced transcription activation at the whole-transcriptome level. Consistent with the ChIP-seq data, data from RNA-seq analysis revealed that M2 gene expression is enhanced

and enriched in IL-4-stimulated *Trim24*-deficient macrophages through gene set enrichment analysis (GSEA) (Fig. 6a, b). Moreover, QPCR and immunoblot assays confirmed that Trim24 negatively regulated M2 gene expression at both the mRNA and protein levels in primary murine macrophages (Fig. 6c). In contrast, the levels of TLR ligands and/or IFNγ-induced expression of proinflammatory genes and the activation of NF-κB and MAP kinases were comparable between WT and *Trim24*-deficient macrophages (Supplementary Fig. 5). These data

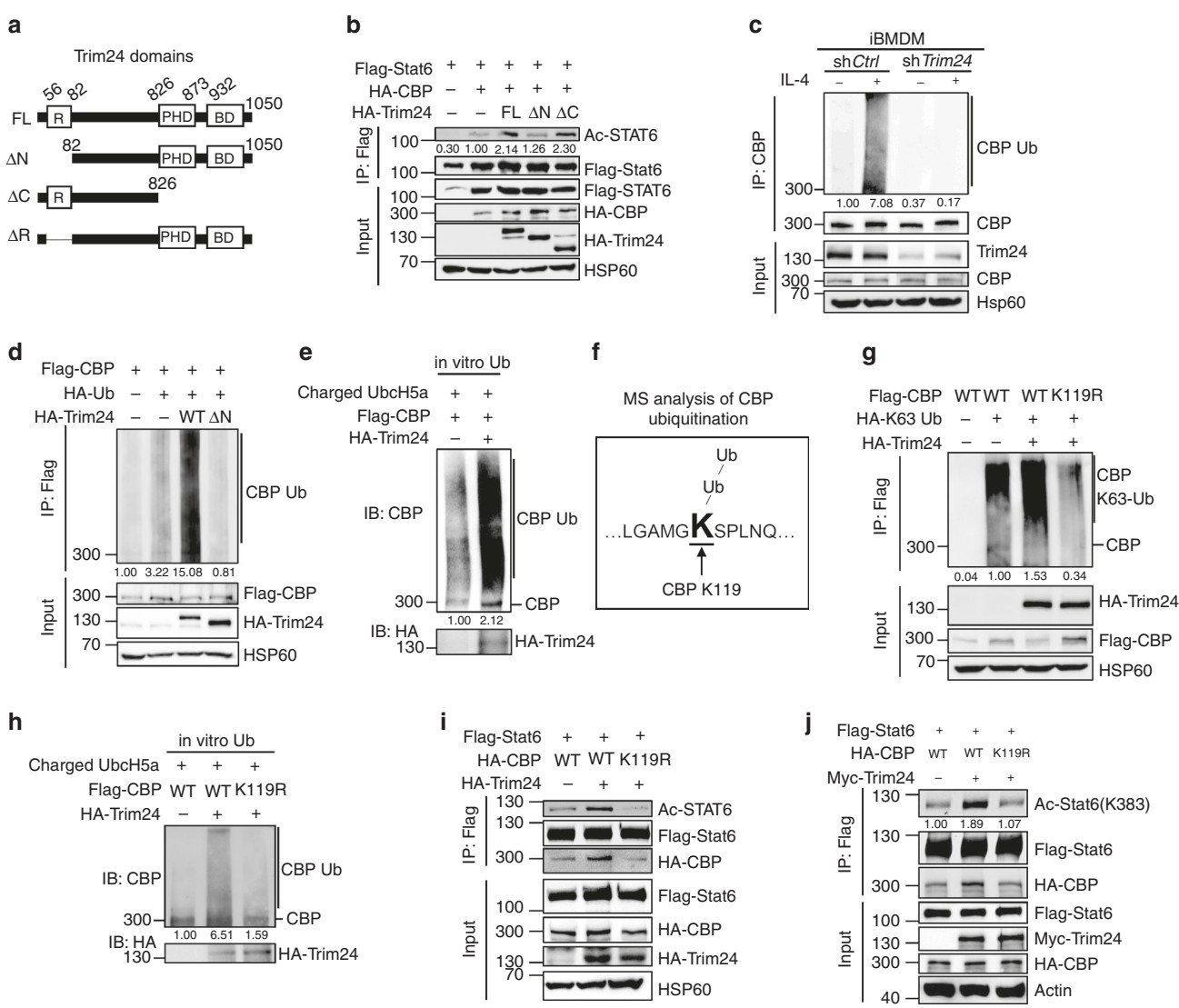

**Fig. 4** Trim24-mediated Lys63-linked ubiquitination of CBP at Lys119. **a** Structure schematic of full-length (FL) Trim24 and its truncations. **b** Immunoblot analysis of the lysine acetylation of Stat6 in 293T cells transfected with the indicated expression vectors assessed by immunoprecipitation (IP) with anti-Flag and immunoblotting with anti-Ac-Lys and anti-Flag; lysates were immunoblotted to detect input proteins and loading controls. **c** Endogenous ubiquitination of CBP in control and *Trim24*-knockdown iBMDMs that were unstimulated (−) or stimulated (+) with IL-4 for 30 min assessed by immunoblot analysis with anti-ubiquitin after immunoprecipitation with anti-CBP (top) and immunoblot analysis with input proteins and loading controls (below). **d** Ubiquitination of CBP in 293T cells transfected with the indicated expression vectors assessed by immunoblot analysis with anti-HA after immunoprecipitation with anti-Flag (top) or immunoblot analysis with input proteins in lysates without immunoprecipitation (below). **e** In vitro ubiquitination assay to determine CBP ubiquitination after a mixture reaction of ubiquitin-charged E2 (UbcH5a) and in vitro-translated Flag-CBP with or without HA-Trim24 proteins was assessed by immunoblot analysis with anti-CBP and anti-HA. **f** Mass spectrometry analysis of the CBP ubiquitination site after the immunoprecipitation of overexpressed Flag-CBP in 293T cells transfected with Flag-CBP and HA-Trim24 showing that Lys119 is the site of CBP ubiquitination by Trim24. **g** Ubiquitination of CBP in 293T cells transfected with the indicated expression vectors assessed by immunoblot analysis with anti-HA after immunoprecipitation with anti-Flag (top) or immunoblot analysis with input proteins in lysates without immunoprecipitation (below). **h** In vitro ubiquitination assay to determine CBP ubiquitination after a mixture reaction of ubiquitin-charged E2 (UbcH5a) and in vitro-translated wild-type (WT) or the K119R mutant Flag-CBP with or without HA-Trim24 proteins was assessed by immunoblot analysis with anti-CBP and anti-HA. **i, j** Immunoblot analysis of the lysine acetylation of Stat6 and the interaction of wild-type (WT) or the K119R mutant CBP with Stat6 in 293T cells transfected with the indicated expression vectors, assessed by immunoprecipitation with anti-Flag and immunoblot with anti-Ac-Lys, anti-Ac-Stat6 (K383), anti-HA, and anti-Flag. Each panel is a representative experiment of at least three independent biological replicates. Source data are provided as a Source Data file

suggested that Trim24 functions as a negative regulator of macrophage M2 polarization but is dispensable for M1 polarization.

Next, we investigated whether Stat6 is required for the enhanced M2 polarization in *Trim24*-deficient macrophages and examined activation of the IL-4-induced signaling pathway. Trim24 deficiency neither affected IL-4-induced phosphorylation of Stat6 and Akt nor promoted the translocation of Stat6 into the nucleus (Supplementary Fig. 6a, b). However, treatment with tofacitinib, a Jak3-selective inhibitor, dramatically suppressed M2 gene induction and abolished the difference in M2 gene expression between WT and *Trim24*-deficient cells (Supplementary Fig. 6c). In addition, *Stat6* knockdown in mouse immortalized bone marrow-derived macrophages (iBMDMs) exerted an

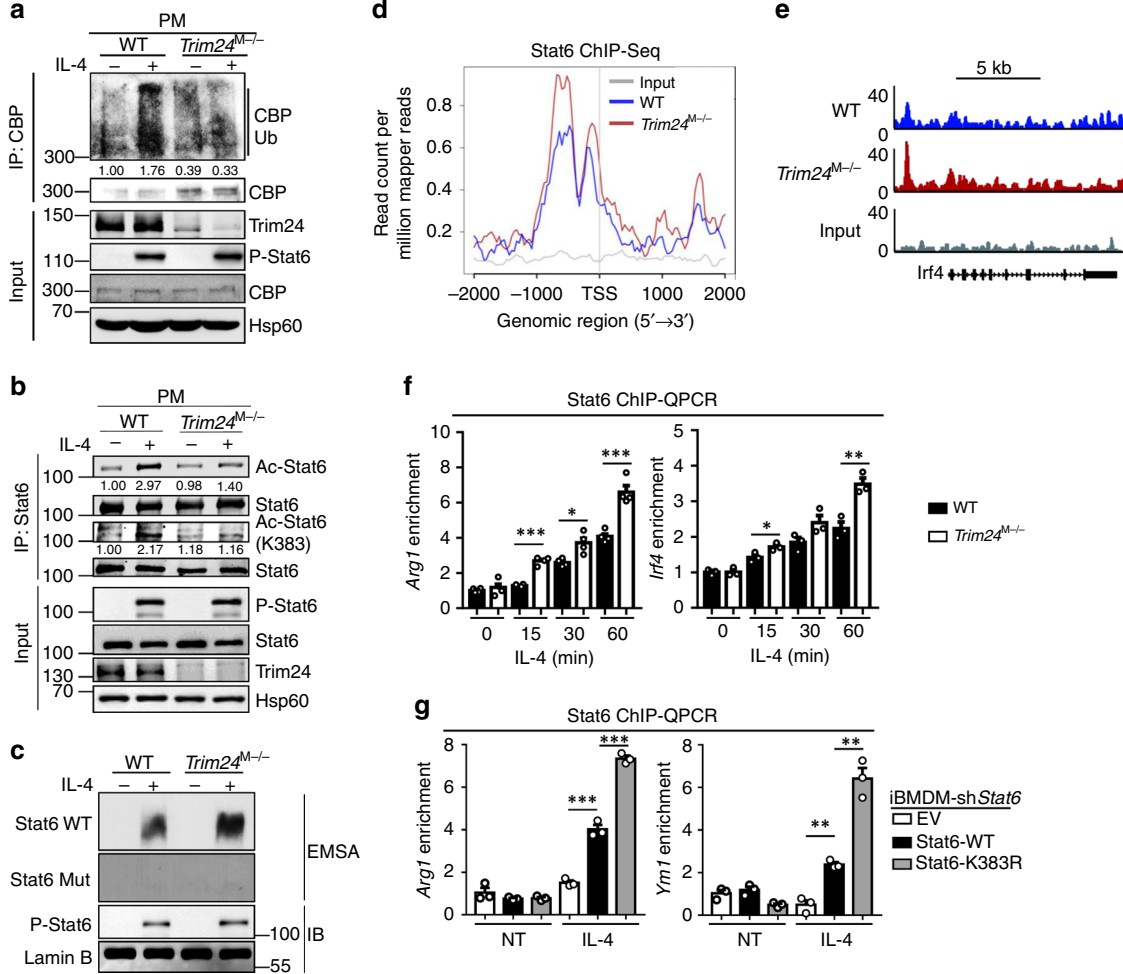

**Fig. 5** *Trim24* deficiency enhanced DNA-binding activity of Stat6 in M2 gene promoters. **a** Endogenous ubiquitination of CBP in WT and *Trim24*-deficient macrophages that were unstimulated (−) or stimulated (+) with IL-4 for 30 min assessed by immunoblot analysis with anti-ubiquitin after immunoprecipitation with anti-CBP (top) and immunoblot analysis with input proteins and loading controls (below). **b** Immunoblot analysis of endogenous lysine acetylation of Stat6 in WT and *Trim24*-deficient peritoneal macrophages (PMs) that were unstimulated (−) or stimulated (+) with IL-4 for 30 min assessed by immunoprecipitation (IP) with anti-Stat6 and immunoblotting with anti-Ac-Lys, anti-Ac-Stat6 (K383), and anti-Stat6. **c** Electrophoretic mobility shift assay (EMSA) of nuclear extracts from WT and *Trim24*-deficient macrophages that were untreated (−) or stimulated (+) with IL-4 for 1 h assessed with $^{32}$P-radiolabeled WT or mutant probes for Stat6. Immunoblotting of phosphorylated (P−) Stat6 and Lamin B in the nuclear extract were applied for control. **d** Average Stat6 ChIP signals in the promoter regions of M2 cluster genes in WT and *Trim24*-deficient macrophages stimulated with IL-4 for 1 h. **e** Snapshot of the Stat6 ChIP-Seq signals at the *Irf4* gene loci in WT and *Trim24*-deficient macrophages stimulated with IL-4 for 1 h. **f**, **g** ChIP–QPCR analysis of Stat6 binding to the promoters of *Arg1* and *Irf4* in WT and *Trim24*-deficient macrophages that were left nontreated or stimulated with IL-4 for the indicated time points (**f**) or Stat6 binding to the promoters of *Arg1* and *Ym1* in *Stat6*-knockdown iBMDMs that were reconstituted with empty vector (EV), WT, or the K383R mutant Stat6 and then were left nontreated (NT) or stimulated with IL-4 for 1 h (**g**). Data with error bars are represented as mean ± SD. Each panel is a representative experiment of at least three independent biological replicates. *$p < 0.05$, **$p < 0.01$, ***$p < 0.001$ as determined by the unpaired Student's *t* test. Source data are provided as a Source Data file

effect similar to that of tofacitinib that was characterized by suppressed and comparable M2 gene induction in both *Trim24*-knockdown and control cells (Supplementary Fig. 6d, e). These results suggested that Stat6 is indispensable for the upregulation of M2 genes in *Trim24*-deficient macrophages, although IL-4-induced phosphorylation and the nuclear translocation of Stat6 were not affected. More interestingly, *TRIM24* knockdown in human peripheral blood mononuclear cell (PBMC)-derived primary macrophages significantly inhibited IL-4-induced Stat6 acetylation at Lys383 and dramatically enhanced the expression of M2 genes accordingly (Fig. 6d–f). These data collectively suggested that Trim24-mediated Stat6 K383 acetylation curtails Stat6–DNA association and thus negatively regulates macrophage M2 polarization.

**Macrophage Trim24 potentiated antitumor immunity**. M2-like TAMs exhibit an immunosuppressive function and thus impair antitumor immunity in the tumor niche[4–6]. Our finding that Trim24 negatively regulates macrophage M2 polarization raised the intriguing question of whether Trim24 modulates antitumor immunity in TAMs. We produced an animal model through the subcutaneous inoculation of B16 melanoma cells to study the in vivo antitumor immune function of Trim24 in macrophages. The loss of Trim24 in myeloid cells dramatically promoted tumor growth (Fig. 7a–c). Interestingly, after collecting TAMs from WT and *Trim24*$^{M−/−}$ tumor-bearing mice through FACS sorting, much more Lys383-acetylated Stat6 was detected by immuno-fluorescent staining in the nuclei of WT TAMs than those of *Trim24*-deficient TAMs (Fig. 7d). Consistently, flow cytometric

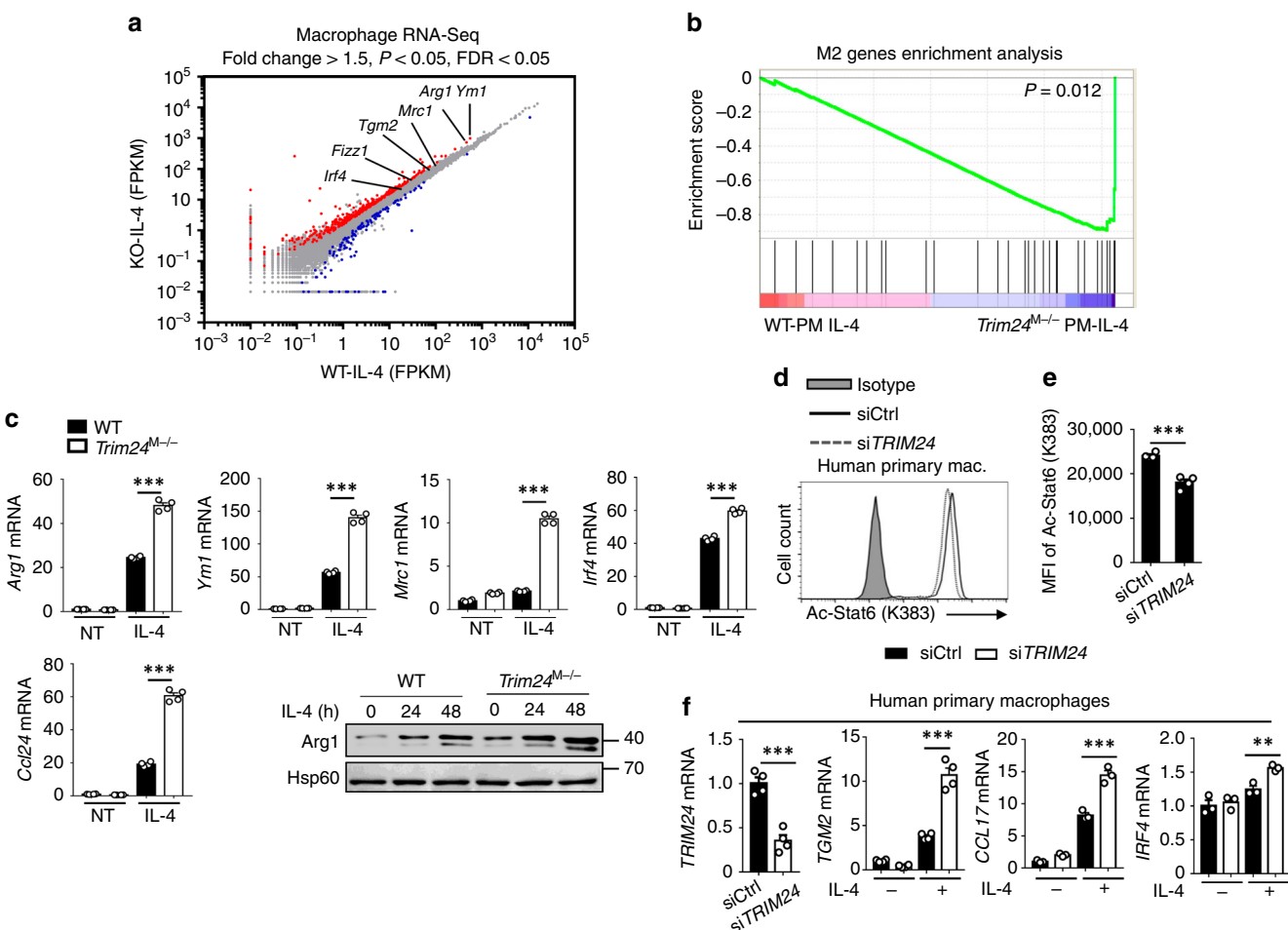

**Fig. 6** Trim24 negatively regulated macrophage M2 polarization. **a** RNA-sequencing analysis showing differentiated gene expression profiles in WT and *Trim24*-deficient macrophages stimulated with IL-4 for 6 h. Representative M2 genes that were upregulated in *Trim24*-deficient macrophages are indicated in the scatter plot. **b** Gene set enrichment analysis (GSEA) of the M2 gene set in WT and *Trim24*-deficient macrophages stimulated with IL-4 for 6 h by using the RNA-sequencing data from (**a**). **c** QPCR and immunoblot analysis of M2 gene expression in WT and *Trim24*-deficient macrophages that were left nontreated (NT) or stimulated with IL-4. **d**, **e** Flow cytometric analysis of endogenous Stat6 acetylation at Lys383 in control and *TRIM24*-knockdown human peripheral blood mononuclear cell (PBMC)-derived primary macrophages that were stimulated with IL-4 for 30 min. Data are presented as a representative histogram (**d**) and summary bar graph (**e**). **f** QPCR analysis of *TRIM24*, *TGM2*, *CCL17*, and *IRF4* mRNA in control and *TRIM24*-knockdown human PBMC-derived primary macrophages that were left nontreated (−) or stimulated (+) with IL-4 for 6 h. Data with error bars are represented as mean ± SD. Each panel is a representative experiment of at least three independent biological replicates. *$p < 0.05$, **$p < 0.01$, ***$p < 0.001$ as determined by the unpaired Student's *t* test. Source data are provided as a Source Data file

analysis confirmed that *Trim24* deficiency dramatically inhibited endogenous Stat6 acetylation at Lys383 in TAMs (Fig. 7e, f). In addition, *Trim24*$^{M-/-}$ mice in a parallel experiment also displayed enhanced tumor growth induced by MC-38 colon adenocarcinoma cells (Fig. 7g). As expected, Trim24 ablation significantly promoted the expression of M2 signature genes in TAMs, including *Arg1*, *Ym1*, *Mrc1*, and *Fizz1*, and the macrophage-recruiting chemokine *Ccl24* (Fig. 7h–j). Consistently, the *Trim24*$^{M-/-}$ mice displayed notably increased tumor infiltration of CD11b$^+$F4/80$^+$ macrophages (Fig. 7k), whereas the infiltration of myeloid-derived suppressor cells (MDSCs) and dendritic cells was not affected by Trim24 ablation (Supplementary Fig. 7a). Accordingly, the myeloid cell's Trim24 deficiency impaired the tumor infiltration of IFNγ-producing CD4$^+$ and CD8$^+$ effector T cells (Fig. 7l), suggesting that *Trim24* deficiency impaired antitumor immunity through suppressed Stat6 acetylation in TAMs.

Since LysM cre is primarily expressed in myeloid cells, including macrophages and MDSCs, we next examined whether Trim24 regulates antitumor immunity in macrophages or MDSCs. To that end, we specifically deleted macrophages or MDSCs by injecting anti-CSF1R or anti-Ly-6G antibodies and challenged mice with B16 melanoma cells. Macrophage deletion significantly suppressed tumor growth in *Trim24*$^{M-/-}$ mice and abolished the difference in the tumor growth rate between WT and *Trim24*$^{M-/-}$ mice (Fig. 7m, n, Supplementary Fig. 7b, c). In contrast, deletion of MDSCs did not perturb the difference in tumor growth between WT and *Trim24*$^{M-/-}$ mice (Supplementary Fig. 7d–g). These data therefore collectively established the essential role of Trim24-modulated Stat6 acetylation in potentiating antitumor immunity in macrophages.

**Stat6 directly suppressed macrophage *Trim24* expression.** High levels of IL-4 in the tumor niche drive the M2 polarization of TAMs and thus impair antitumor immunity. However, our data suggested that IL-4-induced Stat6 acetylation functioned as a negative feedback loop to restrain M2 polarization in macrophages. This discrepancy prompted us to speculate that some

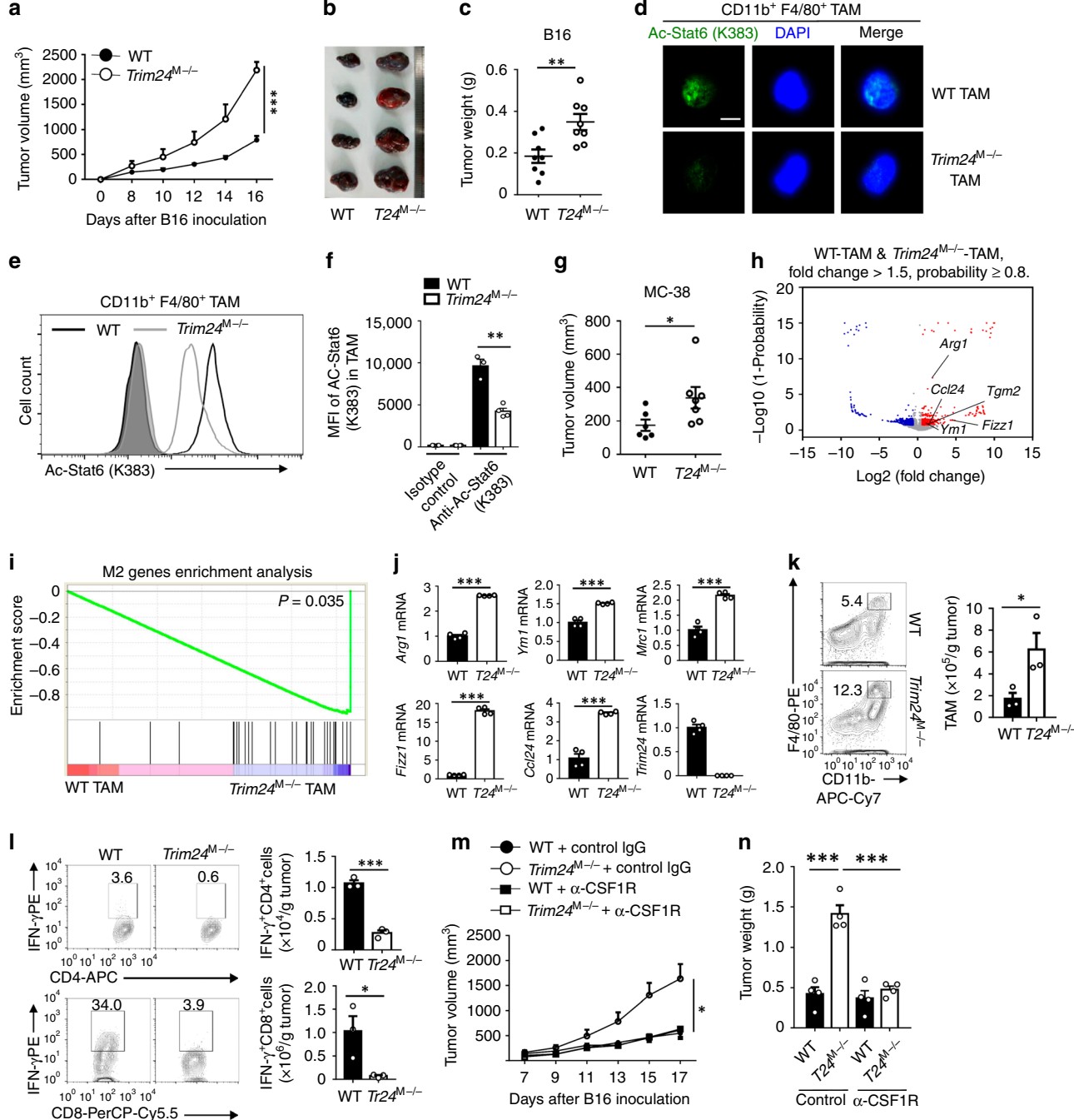

**Fig. 7** Trim24 deficiency in macrophages impaired antitumor immunity. **a–c** Tumor growth (**a**), representative tumor pictures (**b**), and tumor weight (**c**) of WT and *Trim24*M−/− mice (*n* = 8) that were injected s.c. with B16 melanoma cells. **d** Confocal microscopy images showing Lys383-acetylated Stat6 in the nucleus of tumor-associated macrophages (TAMs) that isolated from B16 melanoma of WT and *Trim24*-deficient mice; DAPI staining indicated the nucleus of the cells. Scale bars: 5 μm. **e**, **f** Flow cytometric analysis of endogenous Stat6 acetylation at Lys383 in WT and *Trim24*-deficient TAM as described in (**d**). Data are presented as a representative histogram (**f**) and summary bar graph (**f**). **g** Tumor weight of WT (*n* = 6) and *Trim24*M−/− mice (*n* = 7) that were injected s.c. with MC-38 colon adenocarcinoma cells. **h–j** RNA-sequencing analysis of gene profiles (**h**), M2 gene enrichment (**i**), and QPCR verification of M2 genes (**j**) in WT and *Trim24*M−/− TAM that were isolated from B16 melanoma. The M2 signature genes were indicated in the plot. **k**, **l** Flow cytometric analysis of CD45+CD11b+F4/80+ TAM (**k**) and IFNγ-producing CD4+ and CD8+ T cells (**l**) in B16 melanoma of WT and *Trim24*M−/− mice. Data are presented as a representative plot (left panel) and summary graph (right panel). **m**, **n** Tumor growth (**m**) and weight (**n**) in WT or *Trim24*M−/− mice (*n* = 4) that were injected s.c. with B16 melanoma cells, and with intravenous injection of the anti-CSF1R antibody to deplete macrophages in vivo. Data with error bars are represented as mean ± SD. Each panel is a representative experiment of at least three independent biological replicates. *\*p* < 0.05, *\*\*p* < 0.01, *\*\*\*p* < 0.001 as determined by the unpaired Student's *t* test. Source data are provided as a Source Data file

factors may counteract the effect of Stat6 acetylation in TAMs. To this end, we collected CD11b$^+$ macrophages from breast tumor and adjacent normal tissues from the same cancer patient and examined expression of the *TRIM24* and M2 genes in six patients. Decreased *TRIM24* expression and increased expression of the M2 genes *CCL17* and *IRF4* were detected in macrophages isolated from breast tumors compared with their levels in adjacent normal tissues (Fig. 8a). Therefore, we speculated that Trim24 in macrophages is downregulated by the sensing of environmental factors during tumor pathogenesis. Indeed, both the culture supernatant of B16 melanoma cells and the tumor signature Th2 cytokine IL-4 significantly inhibited *Trim24* expression in both mouse and human macrophages (Fig. 8b–e).

To investigate how *TRIM24* expression is suppressed, we generated luciferase reporter plasmids by introducing different human *TRIM24* promoter fragments into the pGL4-basic plasmid. Interestingly, the promoter region between −1167 and −380 dramatically promoted luciferase activity compared with the pGL4-basic reporter (Fig. 8f, g), suggesting that this promoter fragment is the major region responsible for transcriptional activation of the *TRIM24* gene. A recent study suggested that the transcription factor Stat6 directly induces both transcriptional

activation and suppression in macrophages[30], so we speculated that IL-4-induced Stat6 directly mediates the transcriptional repression of *TRIM24* in M2-polarized macrophages. Indeed, Stat6 deficiency abolished the IL-4-induced suppression of Trim24 expression in macrophages (Fig. 8h). In addition, a conserved Stat6-binding motif (TTCN$_4$GAA) in the *TRIM24* gene promoter region between −1167 and −380 was identified (Fig. 8i), and ChIP–QPCR analysis confirmed that IL-4-induced activated Stat6 could directly bind to this promoter region (Fig. 8j). Moreover, IL-4 stimulation dramatically suppressed the luciferase activity driven by this promoter fragment, and Stat6 overexpression further enhanced the suppressive effect of *TRIM24* transcriptional activity (Fig. 8k). Taken together, these data showed that IL-4-activated Stat6 directly suppresses *TRIM24* gene expression during macrophage M2 polarization, which may in turn contribute to immunosuppressive profiles in the context of the tumor microenvironment.

## Discussion

Stat6 is the major transcription factor responsible for the induction of M2 genes during macrophage M2 polarization[14]. Considering the critical negative regulatory role of M2-prone

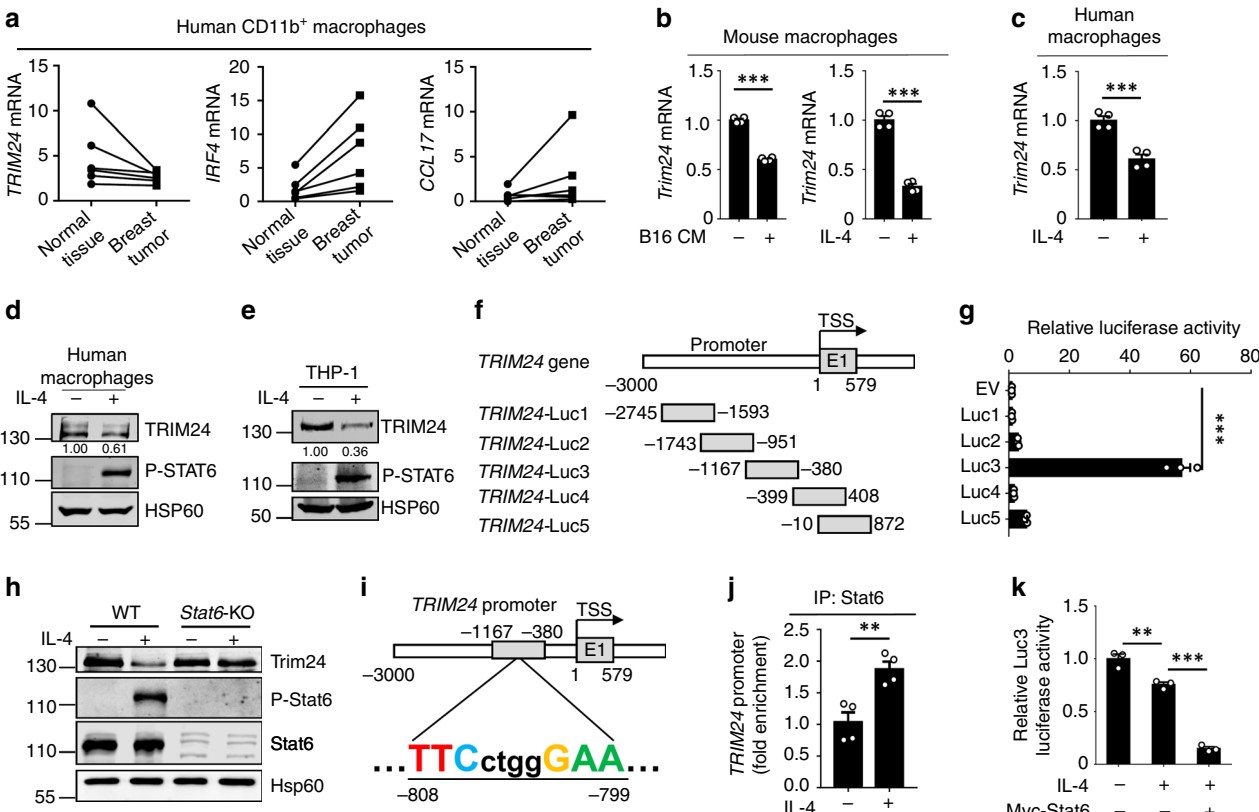

**Fig. 8** Stat6-mediated direct transcriptional suppression of Trim24. **a** QPCR analysis of *TRIM24*, *IRF4*, or *CCL17* mRNA expression in sorted CD11b$^+$ macrophages that isolated from six pairs of breast tumor and adjacent normal tissues from six breast cancer patients. **b**–**e** QPCR and immunoblot analysis of *Trim24* gene expression in murine primary macrophages or in human PBMC-derived macrophages or in human THP-1 cells that were left nontreated (−) or stimulated (+) with a conditioned medium of B16 melanoma cells or IL-4 for 6 h. **f**, **g** Structure schema of the constructed luciferase reporters by using different truncated promoter sequences of *TRIM24* gene, which were then used to test the luciferase activity. **h** Immunoblot of Trim24, phosphorylated (P), and total Stat6 and Hsp60 (loading control) expression in macrophages isolated from wild-type (WT) and *Stat6*-deficient (*Stat6*-KO) mice. **i** Schematic representation showing a Stat6-conserved binding site located in the promoter region of *TRIM24* gene between −1167 and −380. **j** ChIP–QPCR analysis of the binding of STAT6 in the promoter region of *TRIM24* gene in the THP-1 cells that were stimulated with (+) or without (−) IL-4 for 1 h. **k** Luciferase assay of Stat6-induced transcriptional suppression of *TRIM24* in 293T cells that were transfected with the indicated expression vector, and then were stimulated with (+) or without (−) IL-4 for 4 h. Data with error bars are represented as mean ± SD. Each panel is a representative experiment of at least three independent biological replicates. *$p < 0.05$, **$p < 0.01$, ***$p < 0.001$ as determined by the unpaired Student's *t* test. Source data are provided as a Source Data file

TAMs in regulating antitumor immunity[4–6], the study of Stat6 transcriptional modulation may promote the identification of therapeutic targets for cancer immunotherapy. The present study showed that in healthy conditions with high levels of Trim24, the Th2 cytokine IL-4 can induce Stat6 acetylation by ubiquitinated CBP, which is mediated by its association with Trim24. Stat6 acetylation compromised the transcriptional activity of M2 genes and thus restrained macrophage M2 polarization and potentiated antitumor immunity. However, in the tumor microenvironment, activated Stat6 could directly mediate the transcriptional suppression of *Trim24* in M2-polarized macrophages, in which low levels of Trim24 were insufficient to mediate CBP ubiquitination and Stat6 acetylation. In this case, Stat6-mediated Trim24 downregulation further promoted M2 polarization and impaired antitumor immune function, forming a positive feedback loop to contribute to the immunosuppressive protumor niche (Supplementary Fig. 8). Here, we identified the first acetylation modification in Stat6 to curtail its transcriptional activity, through which macrophage M2 polarization and antitumor immunity were controlled.

Protein lysine acetylation, a dynamic process in which an acetyl group is introduced to a specific lysine through an acetyltransferase, such as CBP, p300, and PCAF, has critical functions in multiple biological processes through modulating transcriptional activity and stabilizing specific substrate proteins[31–33]. The regulation of acetylation in histone proteins and their functions have been well characterized in published studies, which show that acetylation is a key epigenetic modulatory mechanism to control gene transcription[34,35]. Nevertheless, increasing research has shed light on the role of lysine acetylation in regulating the functions of transcription factors. The negatively charged phosphate backbone of DNA tends to attract positively charged lysine and arginine residues in the DBDs of transcription factors. However, the acetylation of lysine residues neutralizes their positive charges and thus suppresses the DNA-binding affinity and transcriptional activity of acetylated transcription factors, whereas deacetylation promotes their transcriptional activity[27,28,36]. For example, a recent study suggested that the acetylation of Lys242, Lys245, and Lys262 in Foxo1 inhibits its DNA-binding affinity and thus attenuates its transcriptional activity[28]. In concert with these studies, here, the lysine acetylation of Stat6 indeed inhibited its DNA-binding affinity for the promoters of M2 signature genes and thus suppressed the expression of the corresponding genes.

The acetylation of STAT family proteins has been extensively studied, and several acetylation sites in STAT1, STAT2, STAT3, and STAT5 have been identified[20–24]. A previous study revealed that the HDAC4-induced deacetylation of both Stat6 and histone 3 is important for Arg1 transcription in dendritic cells[37]. The present study also suggested that IL-4-induced Stat6 acetylation negatively regulates the expression of M2 genes in macrophages. However, until now, the exact acetylation site in Stat6 had not been identified; thus, little is known about the biological consequence of its site-specific lysine acetylation at present. A recent structure-based study of Stat6 suggested that K284, K288, K367, and K369 in the DBD of Stat6 are critical for its DNA-binding activity, and Stat6 mutants generated by the replacement of these lysine residues with aspartic acid residues disrupted the association of Stat6 with the corresponding DNA sequence[38]. However, although these lysine residues in Stat6 are critical for its association with DNA, whether these four lysine residues are acetylation sites in Stat6 and the related biological functions of these residues were not known until now. Nevertheless, our study demonstrated that none of these lysine residues is the actual acetylation site of Stat6, and we identified the first functional acetylation site in Stat6, Lys383 in the DBD of Stat6. In addition,

Stat6 acetylation at Lys383 suppressed its DNA-binding activity, thus restraining macrophage M2 polarization, and potentiated the antitumor immune responses of macrophages. Published studies have suggested that *Trim24* deficiency in hepatocytes promotes retinoic acid receptor (RAR)-dependent gene expression[39], and a partner of RAR, retinoid X receptor (RXR), showed widespread overlap with STAT6 binding in human CD14[+] monocyte-derived macrophages[40]. Therefore, it is possible that the *Trim24* deficiency-induced inhibition of Stat6 acetylation functions synergistically with RXR to enhance M2 gene expression in macrophages.

Trim family proteins, which contain a conserved N-terminal Ring domain and diverse C-terminal domains, usually function as E3 ligases that mediate the ubiquitination of different substrates[41–43]. Previous studies have suggested that Trim family proteins play important roles in modulating innate immune signaling[42,43], T-cell differentiation[44], and tumor growth and metastasis[45]. Trim24 is a newly identified E3 ubiquitin ligase, and accumulating evidence has revealed the emerging role of Trim24 in regulating the initiation and development of various kinds of tumors[39,46–48]. Here, we found that Trim24 is a CBP-associated protein that functions as a direct E3 ubiquitin ligase to mediate the Lys63-linked ubiquitination of CBP at Lys119. To the best of our knowledge, Trim24 is the first identified E3 ubiquitin ligase that mediates the Lys63-linked ubiquitination of CBP, and we also identified the first functional Lys63 ubiquitination site in CBP as Lys119. In addition, the Lys119 of CBP is evolutionally conserved with Lys123 of p300. Considering the functional redundancy between CBP and p300 and the Stat6 acetylation that is slightly increased in CBP-knockdown or *Trim24*-deficient macrophages, we speculated that Trim24 may also function as a E3 ligase to mediate p300 ubiquitination, and then promote Stat6 acetylation. Since Trim24 and CBP are primarily located in the nucleus, Trim24-mediated CBP ubiquitination may occur in the nuclei of macrophages upon IL-4 stimulation. As a consequence, IL-4 stimulation promotes Stat6 translocation into the nucleus, where Stat6 is recruited to ubiquitinated CBP and acetylated at Lys383. Since Lys383 is apparently important for the contact of Stat6 with DNA, the association of Stat6 with DNA most likely precludes its acetylation by CBP.

In summary, the present study identified the essential negative regulatory role of Stat6 acetylation mediated by Trim24-induced CBP ubiquitination in macrophage M2 polarization. Due to the critical tumor-promoting function of M2-like TAMs, we propose that targeting Trim24 or downstream Stat6 acetylation has clinical potential in cancer immunotherapy.

## Methods

**Patient samples**. For the analysis of gene expression in CD11b[+] macrophages in breast cancer patients, six primary breast cancer patients, who underwent curative-intent surgery at Comprehensive Breast Health Center, Ruijin Hospital, Shanghai Jiao Tong University School of Medicine, were enrolled in this study. None of the patients had received any preoperative treatment. Specimens were obtained from surgical tissues. Informed consent was obtained from all study subjects prior to their inclusion in this study. The sample collection for this study was reviewed and approved by the Ethics Committee of Ruijin Hospital, Shanghai Jiao Tong University School of Medicine.

**Mice**. *Trim24* floxed mice were generated from Shanghai Research Center for Model Organisms following the procedure as described in Supplementary Fig. 3, and then were backcrossed with C57BL/6 mice for at least for six generations. The C57BL/6 background *Trim24* floxed mice were crossed with lysozyme M (LysM)-Cre mice[49] to produce myeloid cell–conditional Trim24 KO mice (*Trim24*[flox/flox] LysM-Cre, termed *Trim24*[M−/−]). Stat6[+/−] mice under C57BL/6 background (SJ-002828) were purchased from Shanghai Research Center for Model Organisms and were intercrossed to generate Stat6[−/−] (KO) and Stat6[+/+] (WT) mice, which were used in the experiments. In all of the experiments, WT littermate controls were used. All mice were maintained in a specific pathogen-free facility, and all animal experiments were complied with all relevant ethical regulations for animal testing

and research, and were in accordance with protocols approved by the institutional Biomedical Research Ethics Committee, Shanghai Institutes for Biological Sciences, Chinese Academy of Sciences.

**Murine tumor models**. Adherent B16−F10 or MC-38 cells were harvested; after washing twice with phosphate-buffered saline (PBS), single-cell suspensions of $1.2 \times 10^6$ cells were subcutaneously injected into the abdomen of 6–8-week-old age- and sex-matched WT and $Trim24^{M-/-}$ mice. Tumor volumes were measured every 2-day intervals with a caliper and calculated by using the equation: $V = (\text{minor tumor axis})^2 \times (\text{major tumor axis}) \times \pi/6$. For analysis of immune cell infiltration in tumor, mice were killed 12–15 days after B16 inoculation, tumors were weighed, and the infiltrated immune cells were collected for flow cytometric analysis.

**Plasmids, antibodies, and reagents**. Stat6 luciferase plasmid p4 × Stat6-Luc2P (35554) were purchased from Addgene. The pcDNA-CBP was provided by Dr. P. Xu[50]. The cDNA encoding Stat6 (NM_009284.2), Trim24 (NM_145076.4), and its truncations was cloned from the mouse spleen and constructed into the pcDNA vector. For the generation of Stat6 or CBP mutants, point mutations were constructed by site-directed mutagenesis.

Flow cytometric antibodies for mouse CD3 (145-2C11), CD4 (RM4–5), CD8a (53–6.7), CD11b (M1/70), CD11c (N418), CD16/CD32 (93), CD19 (1D3), CD25 (PC61.5), CD44 (IM7), CD45 (30-F11), CD62L (MEL-14), Foxp3 (FJK-16s), B220 (RA3-6B2), F4/80 (BM8), and Ly-6G/Ly6C (RB6-8C5) were from eBioscience. Antibodies for Trim24 (14208-1-AP) and GAPDH (60004-1-1G) were from Proteintech. Antibodies for Trim24 (C-4, sc-271266), Hsp60 (H-1, sc-13115), Lamin B (C-20, sc-6216), p65 (C-20, sc-372), IκBα (C-20, sc-371), p38 (H-147, sc-7149), Erk1 (K-23, sc-94), Stat6 (M-20, sc-981), Irf4 (M-17, sc-6059), Ub (P4D1, sc-8017), c-Myc (9E10, sc-40), and donkey anti-goat IgG (HRP, sc-2020) were from Santa Cruz. Antibodies for Arg1 (9819), Mrc1 (91992), Jnk (9252), Tbk1 (3013), CBP (7389), Stat6 (5397), Acetylated-Lysine (9441), Normal Rabbit IgG (2729), P-Jnk (4668), P-Stat6 (9361), P-Akt (2965, 4060), P-p65 (3033), P-IκBα (2859), P-p38 (9215), P-Stat1 (7649), and P-Tbk1 (5483) were from Cell Signaling. Antibodies for β-actin (A2228) and Flag (A8592) were from Sigma-Aldrich. The antibody for HA (2013819) was from Roche. Antibodies for Lys48-Ubiquitin (05-1307) and Lys63-Ubiquitin (05-1308) were from Merck. Antibodies for Alexa Fluor 594 or Alexa Fluor 488-conjugated rabbit IgG (A21207, A11034), and Alexa Fluor Plus 488-conjugated Mouse IgG (A32723) were from Thermo Fisher Scientific. All of the antibodies for flow cytometry were used at a dilution of 1:100, the primary antibodies for immunofluorescence were used at a dilution of 1:400, and the antibodies for immunoblot or secondary antibodies for immunofluorescence were used at a dilution of 1:1000. The antibody for mouse Lys383-acetylated Stat6 was generated from Shanghai Genomics, and was diluted at 1:50 for flow cytometry or was diluted at 1:200 for immunoblot or immunofluorescence examination.

LPS (L3129) was purchased from Sigma-Aldrich. Protein-A/G magnetic beads (HY-K0202) were from MedChemExpress. FastStart universal SYBR Green master mix (4913914001) was from Roche. Proteinase K (A300491) and DAPI dihydrochloride (A606584) were from Sangon Biotech. Klenow fragment DNA polymerase I (2140A) and PrimeScript RT reagent kit (RR037A) were from Takara. LIVE/DEAD™ fixable violet dead cell stain kit (L34963), TRIzol reagent (15596018), RNase A (8003089), and Lipofectamine 3000 (L3000015) were from Thermo Fisher Scientific. EZ-ChIP chromatin immunoprecipitation kit (17-371) and immobilon Western chemiluminescent HRP substrate (WBKLS0500) were from Millipore. Dual-Luciferase reporter assay system (E1960) and TNT quick coupled transcription/translation systems (L1170) for in vitro protein expression were from Promega. UbcH5a/UBE2D1-ubiquitin charged (human recombinant, E2-800) for in vitro ubiquitination assay was from Boston Biochem. ClonExpress II one step cloning kit (C112) and AxyPrep PCR cleanup kit (AP-PCR-250) were from Vazyme and Axygen, respectively. Recombinant mouse interleukin-4 (CK74), human interleukin-4 (CD03), human macrophage colony-stimulating factor (C417), and human interleukin 10 (CD04) were from Novoprotein. Recombinant mouse IFNγ (315-05) was from Peprotech. Tofacitinib (S2789) and Trichostatin A (TSA, S1045) were purchased from Selleck. NAM, 72340 was from Sigma-Aldrich.

**Macrophage preparation and stimulation**. BMDMs were prepared as previously described[49]. In brief, BM cells isolated from mouse tibia and femur were cultured in 10-cm dishes with DMEM containing 20% FBS and L929 conditional medium. After 4–5 days of culture, the adherent macrophages were detached and seeded into culture plates for further experiments.

For the preparation of peritoneal macrophages, 4% thioglycolate (BD) was intraperitoneally injected into 6–8-week-old WT and $Trim24^{M-/-}$ mice. After 4–5 days, mice were killed, and the peritoneal cavity was lavaged with DMEM medium. The peritoneal cells were collected by centrifugation and seeded in the dish. Macrophages were allowed to adhere for 4 h, washed with fresh medium to remove unattached cells, and incubated overnight.

Human PBMC-derived macrophages were prepared as previously described[51]. Briefly, PBMCs isolated from human peripheral blood of healthy donors were cultured in RPMI-1640 medium with 2 mM L-glutamine, 50 ng mL$^{-1}$ M-CSF, 25 ng mL$^{-1}$ IL-10, and 10% FBS. After monocytes differentiated and proliferated

sufficiently to become confluent, which required around 6 days of culture, macrophages were detached for further experiments.

For the stimulation experiments, macrophages were starved overnight in a medium containing 0.5% FBS before being stimulated with IL-4 (20 ng mL$^{-1}$ for murine macrophages, 50 ng mL$^{-1}$ for human macrophages), LPS (100 ng mL$^{-1}$), or IFNγ (100 ng mL$^{-1}$). Total and subcellular extracts were prepared for immunoblot assays, and total RNA was prepared for quantitative real-time PCR assays.

**Flow cytometry analysis**. Single-cell suspensions were subjected to flow cytometry analyses as previously described[52] by using a Beckman Gallios or BD Aria2 flow cytometer. For Stat6 K383 acetylation analysis, the collected macrophages were fixed with Fixation/Permeabilization Buffer (Invitrogen) on ice for 20 min, washed with Permeabilization Buffer (Invitrogen), and spun down at 3000 rpm for 5 min at 4 °C. Cells were then resuspended and incubated on ice with anti-Ac-Stat6 (K383) or isotype IgG control antibody for 1 h, followed by washing with Permeabilization Buffer, and spun down at 3000 rpm for 5 min at 4 °C. The cell precipitate was resuspended with PE-conjugated anti-rabbit secondary antibody (Invitrogen) in Permeabilization Buffer and incubated on ice for 1 h. After washing with Permeabilization Buffer and spinning down at 3000 rpm for 5 min at 4 °C, the cell was resuspended with permeabilization buffer and analyzed by using a Beckman Gallios or BD Aria II flow cytometer. The flow cytometry data were analyzed by using FlowJo software.

**Real-time quantitative PCR**. Total RNA was extracted by TRIzol reagent according to the manufacturer's protocol. cDNA was synthesized by using the PrimeScript RT Reagent Kit (Takara). QPCR was performed in triplicate by using SYBR Green Master mix (Roche). The relative expression of genes was calculated by a standard curve method and normalized to the expression level of *Actb*. Gene-specific PCR primers are listed in Supplementary Table 1.

**Gene knockdown in mouse and human macrophages**. For knockdown of *Stat6*, *CBP*, or *Trim24* genes in mouse-immortalized macrophages, the pLKO.1 vectors containing shRNA sequences targeting specific genes along with lentiviral-packaging vectors, psPAX2 and pMD2.G[53], were transfected into 293 T cells with Lipofectamine 3000, and after 48 h, we collected the lentiviral supernatants for mouse macrophage infection. The infected cells were selected with puromycin (8 µg mL$^{-1}$) for 48 h and examined the knockdown efficiency by QPCR analysis. For knockdown of *TRIM24* in human PBMC-derived macrophages, siRNAs targeting *TRIM24* (siTRIM24) or negative control (siControl) were transfected into human macrophages with Lipofectamine RNAiMAX. After 48 h, the cells were starved for further experiments. The shRNA and siRNA sequences for targeting specific genes are listed in Supplementary Table 2.

**Immunofluorescence staining**. Cells were fixed for 10 min with 4% cold paraformaldehyde and then were permeabilized for 5 min with 0.2% Triton X-100. After blocking with 2% bovine serum albumin in PBS containing 0.5% Tween-20, cells were stained with specific primary antibodies, followed by blotting with fluorescent-conjugated secondary antibodies. Nuclei were labeled with DAPI (Sangon Biotech). The stained cells were visualized and photographed with ZEISS Cell Observer.

**RNA-sequencing analysis**. Mouse peritoneal macrophages or TAMs that isolated from the B16 melanoma model were applied for total RNA extraction with RNeasy Mini Kit (QIAGEN) and subjected to RNA-sequencing analysis by BGI Tech Solutions. The raw transcriptomic reads were mapped to a reference genome (GRCm38/mm10) by using Bowtie. Gene expression levels were quantified by the RSEM software package. Significantly affected genes were acquired by setting a fold change > 1.5 and a false discovery rate threshold of 0.05.

**In vivo macrophage or MDSC depletion**. For in vivo macrophage or MDSC depletion, 150 µg of the anti-CSF1R-neutralizing antibody (clone AFS98, BioX-Cell), anti-Ly-6G antibody (clone 1A8, BioXCell), or control IgG were intravenously injected every 3 days, with the first injection 24 h before B16 incubation. The tumor growth was monitored and measured as described above. The depletion efficiency for macrophages or MDSC was examined by staining anti-CD45, anti-CD11b, anti-F4/80, or anti-Ly-6G/Ly6C antibodies through flow cytometry.

**Electrophoretic mobility shift assay**. Peritoneal macrophages were stimulated with IL-4 (20 ng mL$^{-1}$) for 1 h, and the nuclear extracts were subjected to EMSA with a $^{32}$P-radiolabeled Stat6-bound WT probe (5′-ATGCTTTCTTATGAACA GGCTG-3′), or mutant probe (5′-ATGCTCGCAGCGACCCAGCCTG-3′).

**Luciferase reporter assay**. Luciferase reporter was co-transfected with pRL-TK and other expression vectors where indicated into 293 T cells by using LipoFiter Transfection Reagent (HanBio). Stat6 transcriptional activity was measured with Dual-Luciferase Reporter Assay System (Promega) and the relative light

unit of chemiluminescence was measured by LB 9508 Lumat3 (Berthold Technologies).

**Co-immunoprecipitation and immunoblot analysis**. For co-immunoprecipitation assays, cells were lysed with RIPA buffer containing protease/phosphatase inhibitors. The whole-cell lysates were incubated with the desired antibodies, and the target protein was then pulled down with protein-A/G magnetic beads. For immunoblot analysis, immunoprecipitates or whole-cell lysates were resolved by using sodium dodecyl sulphate polyacrylamide gel electrophoresis (SDS-PAGE), transferred to nitrocellulose membranes (Millipore), and then blotted with specific primary and secondary antibodies. Immunoblots were visualized by using the immobilon western chemiluminescent HRP substrate (Millipore) with luminescent imaging workstation (Tanon).

**Ubiquitination assay**. For the in vivo ubiquitination assay, macrophages that were left unstimulated or stimulated with IL-4 or 293 T cells that transfected with the desired plasmids were lysed with cell lysis buffer containing protease inhibitor and N-ethylmaleimide. After saving some cell extracts as input analysis, the remaining cell extracts were then boiled at 100 °C for 5 min in the presence of 1% SDS and then diluted with RIPA buffer until the SDS concentration was 0.1%. The diluted lysates were pre-cleaned with Protein-A/G-coupled agarose beads, and then incubated with specific immunoprecipitation antibodies on a shaker at 4 °C overnight. The next day, the immunoprecipitated proteins were collected by incubation with Protein-A/G-coupled agarose beads on a shaker at 4 °C for 2 h, washed with RIPA buffer containing protease inhibitors, PMSF and N-ethylmaleimide, boiled at 100 °C for 5 min, and then loaded to run SDS-PAGE. The immunoprecipitates were immunoblotted with anti-ubiquitin or the indicated antibodies.

For the in vitro ubiquitination assay, CBP, CBP-K119R, and Trim24 proteins were expressed in vitro with the TNT Quick Coupled Transcription/Translation Systems (Promega). In vitro ubiquitination reactions were performed with the Ubiquitination Kit (Boston Biochem) according to the manufacturer's instructions. The reactions were terminated by boiling for 5 min in SDS sample buffer, and were subjected to SDS-PAGE, and followed by immunoblot analysis to examine the ubiquitination of CBP.

**Mass spectrometry**. Flag-Stat6 plus HA-CBP or Flag-CBP plus HA-Trim24 expression plasmids were co-transfected into 293 T cells. Cells were harvested 48 h after transfection, and the lysates were immunoprecipitated with the anti-Flag antibody. After washing, the eluted samples were resolved with SDS-PAGE, followed by Coomassie brilliant blue staining. The sample of Stat6 or CBP band was cut and sent to process with MS analysis by using the QE1 system in Shanghai Science Research Center.

**ChIP assay**. ChIP assay procedure was modified from the manufacturer's instructions (EZ-ChIP, Millipore). Briefly, isolated mouse peritoneal macrophages or human THP-1 cells (about $1 \times 10^7$ cells) were fixed with 1% formaldehyde (Sigma-Aldrich) at room temperature for 10 min in 10-ml media, followed by quenching with 125 mM glycine. Nuclear extracts were sonicated with Covaris E220 for 660 s. After preclearing with normal IgG for 1 h, the sonicated cell lysates were immunoprecipitated with the indicated antibodies overnight on an incubator at 4 °C. The next day, protein-A/G magnetic beads were added, and cell lysates were incubated on an incubator for another 2 h. After washing with buffers, chromatin was eluted from the protein/DNA complex, and digested with proteinase K and RNase A at 65 °C overnight to reverse cross-links. The freed DNA was purified with AxyPrep PCR cleanup kit (Axygen) and subjected to quantitative PCR analysis by using SYGR Green master mix. All the sequences of primers for ChIP–QPCR are shown in Supplementary Table 1.

**Statistical analysis**. The data are shown as mean ± SD, and unless otherwise indicated, all the presented data are the representative results of at least three independent repeats. Statistical analysis was performed by using GraphPad Prism 5 (Graph-Pad Software), and the statistics were analyzed by a two-tailed Student's $t$ test or one-way or two-way ANOVA as indicated. Differences were considered to be significant at $p \le 0.05$ and are indicated by ∗, those at $p \le 0.01$ are indicated by ∗∗, and those at $p \le 0.001$ are indicated by ∗∗∗.

**Reporting summary**. Further information on research design is available in the Nature Research Reporting Summary linked to this article.

## Data availability
The RNA-sequencing and ChIP-sequencing data have been deposited into the Gene Expression Omnibus with the accession codes GSE116588, GSE116566, GSE134158, and GSE134167. The source data underlying Figs. 1a–f, 1i, 2, 3a, c–j, 4b–j, 5a–c, f, g, 6c, e, f, 7a, c, f, g, j–n, 8a–e, g, h, j, k and Supplementary Figs. 2, 3c, 4b, d, f, h, 5, 6, 7a, c–e, g are provided as a Source Data file. All other data supporting the findings of this study are available from the corresponding author upon reasonable request.

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

## Acknowledgements

This research was supported by the grants from the National Key R&D Program of China (2018YFA0902700 and 2018YFA0107201), the Strategic Priority Research Program of the Chinese Academy of Sciences (XDPB10 and XDB19000000), the Key Research Program of the Chinese Academy of Sciences (CAS) (KFZD-SW-216), the National Natural Science Foundation of China (81571545, 81770567, and 81825018), the Jiangsu Provincial Key and Development Program (BE2016722), the Thousand Young Talents Plan of China, and CAS Key Laboratory of Tissue Microenvironment and Tumor.

## Author contributions

T.Y., S.G., and Q.Z. designed and performed the experiments, prepared the figures, and wrote parts of the paper; D.D., H.W., X.C., C.M., and K.S. provided the clinical samples and the related experiments; N.L. contributed to the bioinformatic analysis of RNA-seq and ChIP-seq data; D.H., Y.W., Q.P., J.X., X.Z., J.L., S.P., S.R., M.H., and X.Z. contributed to the experiments; C.P. and P.W. contributed to mass spectrometry analysis; J.Q. contributed to data analysis and critical editing; Y.X. designed and supervised this study, prepared the figures, and wrote the paper.

## Conflict of interest

The authors declare no competing interests.
