## [Peer Review File · Nature Communications]

Reviewers' comments:

Reviewer #1 (Remarks to the Author):

Yu and colleagues described that CBP-mediated acetylation of STAT6 transcription factor attenuates its DNA binding capacity leading to the decreased M2-type macrophage polarization. They also observed that TRIM24 ubiquitin ligase plays key role in this process enhancing the interaction between CBP and STAT6. Finally, the authors demonstrated that the TRIM24-CBP-mediated acetylation of STAT6 regulates the immunosuppressive tumor niche in vivo. These findings strongly suggest that the acetylation of STAT6 is able to modify the IL-4 response in macrophages in vitro and in vivo. Although, the described mechanism is novel and the experimental approaches are appropriate and thorough, but the global analyses of transcriptomic and STAT6 cistromic changes are completely missing. I recommend to complete the manuscript with RNA-seq and STAT6-specific ChIP-seq data. Major points of criticism:

1. The authors suggest that the TRIM24-CBP-mediated acetylation of STAT6 generally attenuates its DNA binding capacity and reduces the expression of the alternative macrophage polarization-specific gene set. However, they showed this phenomenon on some selected genes and genomic regions. I consider it important to prove that TRIM24-CBP axis regulates the IL-4/STAT6 signaling pathway-mediated direct transcription activation at the whole transcriptome level.
2. Some previous publications (Ostuni et al. *Cell*, 2013; Czimmerer et al. *Immunity*, 2018) described that the global IL-4-activated STAT6 binding show strict time dependency. I think it necessary to show with STAT6-specific ChIP-seq method that the IL-4-activated STAT6 binding (or its time dependency) generally affected in macrophages by the inhibition of STAT6 acetylation and K383R mutation.
3. The authors showed on Figure 7A that TRIM24 and TGM2 expressions are significantly regulated in CD11b+ macrophages from malignant tumor compared to benign adenoma-derived macrophages. Are STAT6 and additional alternative macrophage polarization markers (and direct STAT6 target genes) including CCL17, MRC1, IRF4 etc. regulated differently between malignant tumor and benign adenoma-derived CD11b+ macrophages?

Reviewer #2 (Remarks to the Author):

This study reports a novel mechanism of regulation of transcription factor STAT6 DNA binding and activity through acetylation of Lys383 mediated by protein acetyltransferase CBP (KAT3A). The authors discover a novel mechanism of CBP regulation through ubiquitination of Lys119 through ubiquitin E3 ligase Trim24. The authors suggest that Trem24-mediated CBP K63-linked ubiquitination promotes association of CBP and STAT6, leading to Lys383 acetylation of STAT6. This post-translational modification abolishes STAT6 DNA binding and reduces STAT6-mediated, IL-4-induced polarization of murine and human macrophages. Using myeloid-specific Trem24 knockout mice, the authors show that the absence of Trem24 in macrophages promotes tumor growth in tumor xenograft models through increased pro-tumorigenic M2 polarization of macrophages, which the authors attribute to reduced inhibitory STAT6 acetylation. Remarkably, the authors show STAT6-dependent repression of Trem24 expression, which they suggest promotes M2 polarization of macrophages in tumor microenvironment.

In total, the manuscript presents several novel findings of importance towards understanding molecular mechanisms of macrophage polarization and its relation to tumor microenvironment. It also reveals novel function of Trem24 via its interaction with, and ubiquitination of, CBP. These findings definitely have the potential to influence thinking in the field of molecular immunology. However, there are several major conceptual questions which should be clarified before the manuscript, in my view, is acceptable for publication. Furthermore, there are multiple issues regarding the quality and interpretation of multiple experiments, which should be resolved, which I present

below.

Although I consider the manuscript to be unacceptable in its present form, the study seems promising and the authors may consider a resubmission in the future.

1. The authors focus the manuscript on the role of CBP (KAT3A) in STAT6 acetylation. However, this acetyltransferase is highly homologous and in many cases functionally redundant with KAT3B (EP300 or p300) acetyltransferase. Whether p300 can also acetylate STAT6 at Lys383 and thus, substitute for CBP, is an important question which should be addressed by the authors. It is also important to clarify whether p300 can be ubiquitinated by Trem24.

2. Trem24 was initially shown to be an inhibitor of retinoic acid receptor signaling. In particular, transglutaminase 2 was upregulated in Trem24-deficient hepatocytes in a RAR-dependent manner (PMID: 18026104). Since a partner of RAR, RXR, shows widespread overlap with STAT6 binding in macrophages (PMID: 28774779), it would be important to test whether RAR and RXR levels are affected in Trem24 knockout macrophages and whether antagonism of RAR/RXR may prevent upregulation of IL-4 target genes in Trem24 knockout cells.

3. It is not clear how IL-4 signaling promotes STAT6 acetylation. Trem24 and CBP are constitutively nuclear. What should be then the signal promoting Trem2-dependent ubiquitination of CBP? Where and when should CBP acetylate STAT6? Furthermore, upon IL-4-stimulation STAT6 translocates to the nucleus and binds its cognate DNA sequences. Since K383 is apparently important for making contacts of STAT6 with DNA, association of STAT6 with DNA probably disallows its acetylation by CBP. These points should be at least discussed.

4. Throughout the figures, the usage of overexposed western blot images (most of the Flag-STAT6 images) should be avoided, since it masks possible differences in protein amounts. This is especially critical since no western blot quantifications were performed.

5. Language editing should be performed throughout the manuscript.

Specific remarks:

Fig 1B: The enhancement of STAT6 acetylation by IL-4 is not obvious from the blot. Apparently, using overexpression system, adding IL-4 is unnecessary for inducing STAT6 acetylation. I would suggest removing IL-4 from the experiments in 293 cells to avoid confusion.

Fig 1C: The enhancement of STAT6 acetylation in shControl after IL-4 is not obvious since different levels of stat6 are present in the control and IL-4 samples. Although CBP is absent, the increase of Stat6 acetylation in response to IL-4 is still observed – again raising the question whether another acetyltransferase such as p300 also plays a role. For this experiment, please provide the full-length scans of all western blots (at least three biological replicates) with indication of molecular weight marker positions.

Fig 1D: Please indicate concentrations of TSA and NAM used. Please show the effect of CBP overexpression on the reporter activity.

Fig 1E: The drugs have very broad and strong effects on histone acetylation, and therefore it is difficult to attribute their effects on gene expression solely to STAT6 acetylation. I recommend to remove these experiments.

Fig 1H: MS analysis should be presented and described in a way clear to a non-specialist. Were other acetylation sites identified as well?

Fig 2C: The acK383 Antibody is apparently not fully specific, since substantial signal is still present in the mutant – this issue should be discussed.

Fig 2D-E: Detailed description of FACS analysis is necessary. Appropriate controls should be shown. Cell type is not clear. Was the experiment performed in macrophages from tumor? NT should be disabbreviated.

Fig 2F: Why the K383R mutant shows big increase of activity without CBP overexpression? (no acetylation should be expected). Control for protein expression levels of different mutant constructs should be shown.

Fig 2H: pSTAT6 and STAT6 appear sometimes as a doublet here and several other blots. Preferably only one band should be presented to avoid confusion.

Fig 2I: A western blot verifying equal levels of STAT6 expression should be shown as well as the effect of the mutant without IL-4 stimulation.

Lane 163-166: It is unclear how these analyses were conducted – reveal the details and the source of

data.

Fig 3B: For me, Trim24 is partly nuclear independently of IL-4 stimulation. Nuclear fractionation western blot is necessary to make quantitative claims on Trim24 nuclear translocation upon IL-4 stimulation.

Fig 3C: In this experiment, aCLys and Flag images appear to be from different blots. The increase in aCLys upon Trim24 is not obvious. For this experiment, please provide the full-length scans of all western blots (at least three biological replicates) with indication of molecular weight marker positions.

Fig 3D: In this experiment (as well as in some others) Trim24 seems to affect total levels of CBP. The effect of Trim24 on total CBP levels should be addressed.

Fig 3E: Here the promotion of the STAT6-CBP complex by Trim24 is not obvious since: a) CBP levels appear to be also increased by Trim24 OE and b) amount of Flag-STAT6 in the IP seems to be different with and without Trim24 overexpression. For this experiment, please provide the full-length scans of all western blots (at least three biological replicates) with indication of molecular weight marker positions.

Fig 3F: Please show Trim24 levels after knockdown in input samples.

Fig 3G: Please use western blot analysis in this experiment. FACS analysis may be used only if the sample amount is unsuitable for western analysis, which is not the case here.

Fig 3H-I: In these both cases it is unclear how Trim24 achieves effects on STAT6 without overexpressed CBP. Please show the effects on STAT6 acetylation for both overexpression and knockdown of Trim24.

Fig 4C: The effect of IL-4 on CBP Ubiquitination is difficult to judge due to the difference in CBP levels in the IP. For this experiment, please provide the full-length scans of all western blots (at least three biological replicates) with indication of molecular weight marker positions. Please also provide the details of the Anti-Ub western blot experiments in the experimental procedures section.

Fig 4F: Please present the results of the mass-spectrometric analysis in a way clear and understandable to a non-specialist. More details and the description are needed.

Fig 4I: It looks like the amount of Flag-STAT6 in the input is not equal, which makes questionable the effect of Trim24 overexpression on STAT6 acetylation in this experiment.

Fig 4J: Why is HA-Trim24 signal present in the second lane where no HA-Trim24 should be?

Fig 5A: CBP blots both in input and in the IP samples are of very poor quality, please provide better quality blots.

Fig 5B: The fact that IL-4 still enhances STAT6 acetylation in Trim24 knockout macrophages should be discussed.

Fig 5F-G: The usage of TSA/NAM in these experiments should be avoided.

Fig 5H-I: Please use western blot analysis in this experiment. FACS analysis may be used only if the sample amount is unsuitable for western analysis, which is not the case here.

Fig 5J. Please expand the panel of IL-4-dependent genes regulated by Trim24 in human macrophages.

Fig 6J: Ccl24 is reduced in the macrophages, not in the tumor. How this can explain reduction of macrophage infiltration in the tumor?

Fig 6K: Bar chart Y-axis description is the same for two graphs – should lower graph depict the levels of CD8+ cells?

Fig 7D: Why THP-1 and not PBMCs are used in this case?

Point-to-point response letter

Reviewer #1 (Remarks to the Author):

- *Yu and colleagues described that CBP-mediated acetylation of STAT6 transcription factor attenuates its DNA binding capacity leading to the decreased M2-type macrophage polarization. They also observed that TRIM24 ubiquitin ligase plays key role in this process enhancing the interaction between CBP and STAT6. Finally, the authors demonstrated that the TRIM24-CBP-mediated acetylation of STAT6 regulates the immunosuppressive tumor niche in vivo. These findings strongly suggest that the acetylation of STAT6 is able to modify the IL-4 response in macrophages in vitro and in vivo. Although, the described mechanism is novel and the experimental approaches are appropriate and thorough, but the global analyses of transcriptomic and STAT6 cistromic changes are completely missing. I recommend to complete the manuscript with RNA-seq and STAT6-specific ChIP-seq data.*

Response

We thank the reviewer for the positive comments and the insightful suggestions to strengthen our studies. As such, we performed additional experiments suggested by the reviewer as shown below.

Major points of criticism:

1. *The authors suggest that the TRIM24-CBP-mediated acetylation of STAT6 generally attenuates its DNA binding capacity and reduces the expression of the alternative macrophage polarization-specific gene set. However, they showed this phenomenon on some selected genes and genomic regions. I consider it important to prove that TRIM24-CBP axis regulates the IL-4/STAT6 signaling pathway-mediated direct transcription activation at the whole transcriptome level.*

Response

We thank the reviewer for this insightful comment, and therefore performed the RNA-sequencing analysis as suggested. As expected, loss of Trim24 enhanced the expression of M2 cluster genes (revised **Figure 5h**), and gene set enrichment analysis (GSEA) indicated that M2 genes were enriched in IL-4-stimulated *Trim24*-deficient primary cultured macrophages (revised **Figure 5i**) or tumor-associated macrophages (revised **Figure 6i**). Thus, this global transcriptomic analysis supported Trim24-CBP axis indeed regulates the STAT6-mediated direct transcription activation. Moreover, we also confirmed the enhanced expression of M2 genes, such as *Arg1*, *Ym1*, *Mrc1*, etc., in *Trim24*-deficient primary cultured macrophages upon IL-4 stimulation (revised **Figure 5j**) and tumor-associated macrophages (revised **Figure 6j**) by using quantitative RT-PCR analysis. These results collectively showed Trim24-CBP axis indeed controls IL-4/STAT6-mediated transcription activation.

2. *Some previous publications (Ostuni et al. Cell, 2013; Czimmerer et al. Immunity, 2018) described that the global IL-4-activated STAT6 binding show strict time dependency. I think it necessary to show with STAT6-specific ChIP-seq method that the IL-4-activated STAT6 binding (or its time dependency) generally affected in macrophages by the inhibition of*

STAT6 acetylation and K383R mutation.

Response

We thank the reviewer for this constructive comment to improve our manuscript. As suggested, we immunoprecipitated Stat6 from IL-4-stimulated WT and *Trim24*-deficient murine primary macrophages and analyzed the co-immunoprecipitated DNA with deep sequencing (CHIP-Seq). The results revealed that loss of *Trim24*, which suppressed Stat6 K383 acetylation, enhanced IL-4-induced Stat6 binding activity in M2 gene promoters (revised **Figure 5d and 5e**). In addition, per your comments, we also performed time course-dependent ChIP-qPCR, and showed *Trim24* deficiency promoted STAT6 binding activity to M2 gene promoters upon IL-4 stimulation (revised **Figure 5f**). Likewise, K383R mutation in Stat6, which abolished its acetylation, dramatically promoted its binding in M2 gene promoters as compared to WT Stat6 in murine macrophages upon IL-4 stimulation (revised **Figure 5g**). These results collectively indicated that Stat6 K383 is critical for its DNA-binding activity in M2 gene promoters.

3. *The authors showed on Figure 7A that TRIM24 and TGM2 expressions are significantly regulated in CD11b+ macrophages from malignant tumor compared to benign adenoma-derived macrophages. Are STAT6 and additional alternative macrophage polarization markers (and direct STAT6 target genes) including CCL17, MRC1, IRF4 etc. regulated differently between malignant tumor and benign adenoma-derived CD11b+ macrophages?*

Response

We thank the reviewer for the comment. Since the samples that collected from colon cancer have been run out, we collected CD11b⁺ macrophages from the breast tumor and their adjacent normal counterparts, and examined the expression of *TRIM24* and M2 genes from total 6 patients. The results revealed that lower *TRIM24* expression along with higher M2 genes' expression, including *CCL17* and *IRF4*, were detected in macrophages that isolated from breast tumors as compared to that from adjacent normal tissues (revised **Figure 7a**). These new data have been added in the revised figure 7 to replace the results from colon cancer.

Reviewer #2 (Remarks to the Author):

- *This study reports a novel mechanism of regulation of transcription factor STAT6 DNA binding and activity through acetylation of Lys383 mediated by protein acetyltransferase CBP (KAT3A). The authors discover a novel mechanism of CBP regulation through ubiquitination of Lys119 through ubiquitin E3 ligase Trim24. The authors suggest that Trem24-mediated CBP K63-linked ubiquitination promotes association of CBP and STAT6, leading to Lys383 acetylation of STAT6. This post-translational modification abolishes STAT6 DNA binding and reduces STAT6-mediated, IL-4-induced polarization of murine and human macrophages. Using myeloid-specific Trem24 knockout mice, the authors show that the absence of Trem24 in macrophages promotes tumor growth in tumor xenograft models through increased pro-tumorigenic M2 polarization of macrophages, which the authors attribute to reduced inhibitory STAT6 acetylation. Remarkably, the authors show STAT6-dependent repression of Trem24 expression, which they suggest promotes M2 polarization of macrophages in tumor microenvironment.*

In total, the manuscript presents several novel findings of importance towards understanding molecular mechanisms of macrophage polarization and its relation to tumor microenvironment. It also reveals novel function of Trem24 via its interaction with, and ubiquitination of, CBP. These findings definitely have the potential to influence thinking in the field of molecular immunology. However, there are several major conceptual questions which should be clarified before the manuscript, in my view, is acceptable for publication. Furthermore, there are multiple issues regarding the quality and interpretation of multiple experiments, which should be resolved, which I present below. Although I consider the manuscript to be unacceptable in its present form, the study seems promising and the authors may consider a resubmission in the future.

Response

We thank the reviewer for noting that our study is of interest, and for bringing up the insightful comments/suggestions to strengthen our studies. As suggested, we performed the additional experiments and revised the manuscript to improve the clarity.

1. *The authors focus the manuscript on the role of CBP (KAT3A) in STAT6 acetylation. However, this acetyltransferase is highly homologous and in many cases functionally redundant with KAT3B (EP300 or p300) acetyltransferase. Whether p300 can also acetylate STAT6 at Lys383 and thus, substitute for CBP, is important question which should be addressed by the authors. It is also important to clarify whether p300 can be ubiquitinated by Trem24.*

Response

We thank the reviewer for the comment. We and the editor all agree fully addressing this issue will advance our studies; however, it might be out of scope for the present study. Actually, our data have showed that Stat6 acetylation is slightly increased in CBP-knockdown iBMDM after IL-4 treatment (revised **Figure 1c**) and the Lys383 acetylation of Stat6 is also slightly induced in *Trim24*-deficient macrophages upon IL-4 stimulation (revised **Figure 5b**), implying that other acetyltransferases like p300 may substitute CBP for Stat6 acetylation when considering their redundant function as previously described.

In addition, we found that Lys119, the ubiquitination site of CBP identified by mass spectrum analysis, is evolutionally conserved with Lys123 of p300 as showed in the following figure. Considering the functionally redundancy between CBP and p300 and the Stat6 acetylation is also slightly increased in CBP-knockdown or *Trim24*-deficient macrophages (revised **Figure 1c, 5b**), we speculated that Trim24 may also function as a E3 ligase to mediate p300 ubiquitination, and then promote Stat6 acetylation. We have added this discussion in the Results and Discussion section of the revised manuscript.

K119
|
CBP: GAMGKSPLNQ
p300: NSMVKSPMAQ

2. *Trem24 was initially shown to be an inhibitor of retinoic acid receptor signaling. In particular, transglutaminase 2 was upregulated in Trem24-deficient hepatocytes in a RAR-dependent manner (PMID: 18026104). Since a partner of RAR, RXR, shows widespread*

overlap with STAT6 binding in macrophages (PMID: 28774779), it would be important to test whether RAR and RXR levels are affected in Trem24 knockout macrophages and whether antagonism of RAR/RXR may prevent upregulation of IL-4 target genes in Trem24 knockout cells.

Response

We thank the reviewer for this comment, which would broaden the scope of the current study. Per editor and your suggestions, this important question has been discussed in the Discussion section of the revised manuscript as follows:

Published studies have suggested that *Trim24* deficiency in hepatocytes promoted retinoic acid receptor (RAR)-dependent gene expression, and a partner of RAR, retinoid X receptors (RXR), showed widespread overlap with STAT6 binding in human CD14⁺ monocyte-derived macrophages. Therefore, it is possible that *Trim24* deficiency-induced inhibition of Stat6 acetylation function synergistically with RXR to enhance M2 genes' expression in macrophages.

- 3. It is not clear how IL-4 signaling promotes STAT6 acetylation. Trem24 and CBP are constitutively nuclear. What should be then the signal promoting Trem2-dependent ubiquitination of CBP? Where and when should CBP acetylate STAT6? Furthermore, upon IL-4-stimulation STAT6 translocates to the nucleus and binds its cognate DNA sequences. Since K383 is apparently important for making contacts of STAT6 with DNA, association of STAT6 with DNA probably disallows its acetylation by CBP. These points should be at least discussed.*

Response

We thank the reviewer for this insightful comment. As suggested, we discussed this point in the Discussion section in the revised manuscript as follows:

Since *Trim24* and CBP are primarily located in the nucleus, *Trim24*-mediated CBP ubiquitination is supposed to occur in the nucleus of macrophages upon IL-4 stimulation. As a consequence, IL-4-induced nuclear accumulation of Stat6 was recruited to the ubiquitinated CBP and then was acetylated at Lys383. Since Lys383 is apparently important for making contacts of Stat6 with DNA, association of Stat6 with DNA probably might interfere its acetylation by CBP.

- 4. Throughout the figures, the usage of overexposed western blot images (most of the Flag-STAT6 images) should be avoided, since it masks possible differences in protein amounts. This is especially critical since no western blot quantifications were performed.*

Response

We thank the reviewer for this comment. As suggested, we replaced overexposed western blot images with short-exposed ones, and we have provided all the original raw data for all the figures including the full-length scans of all western blots with indication of molecular weight marker as suggested. In addition, we also quantified the levels of Stat6 acetylation and CBP ubiquitination and the values were presented below the related western blot images in the revised manuscripts.

- 5. Language editing should be performed throughout the manuscript.*

Response

As suggested, we have sent the manuscript for a substantial language editing by Nature published editor service. We also have carefully gone through the manuscript, corrected several language errors and made substantial language editing throughout the manuscript.

Specific remarks:

- *Fig 1B: The enhancement of STAT6 acetylation by IL-4 is not obvious from the blot. Apparently, using overexpression system, adding IL-4 is unnecessary for inducing STAT6 acetylation. I would suggest removing IL-4 from the experiments in 293 cells to avoid confusion.*

Response

We have removed IL-4 stimulation group in this experiment as suggested.

- *Fig 1C: The Enhancement of STAT6 acetylation in shControl after IL-4 is not obvious since different levels of stat6 are present in the control and IL-4 samples. Although CBP is absent, the increase of Stat6 acetylation in response to IL-4 is still observed – again raising the question whether another acetyltransferase such as p300 also plays a role. For this experiment, please provide the full-length scans of all western blots (at least three biological replicates) with indication of molecular weight marker positions.*

Response

We thank the reviewer for this comment. After performing a sequence alignment, we observed that the Lys119 of CBP is evolutionally conserved with Lys122 of p300. Considering the functionally redundancy between CBP and p300 and the Stat6 acetylation is also slightly increased in CBP-knockdown or *Trim24*-deficient macrophages (revised **Figure 1c, 5b**), we speculated that *Trim24* may also function as a E3 ligase to mediate p300 ubiquitination, and then promote Stat6 acetylation. We have added this discussion in the Results section of the revised manuscript.

In addition, we have provided all the original raw data for all the figures including the full-length scans of all western blots with indication of molecular weight marker as suggested. The data of three biological replicates for this experiment are as follows:

- *Fig 1D: Please indicate concentrations of TSA and NAM used. Please show the effect of CBP overexpression on the reporter activity.*

Response

As suggested, the concentration of TSA (5 μ M) and NAM (1 mM) were indicated in the related figure legend of the revised manuscript. For the effect of CBP overexpression on the Stat6-driven luciferase activity, we have showed the data in **Figure 2f, Supplementary Fig. 2b**, which suggested that CBP overexpression alone could not induce the report activity.

- *Fig 1E: The drugs have very broad and strong effects on histone acetylation, and therefore it is difficult to attribute their effects on gene expression solely to STAT6 acetylation. I recommend to remove these experiments.*

Response

We have removed this result in the revised manuscript as suggested.

- *Fig 1H: MS analysis should be presented and described in a way clear to a non-specialist. Were other acetylation sites identified as well?*

Response

We apologize for not describing clearly for the MS data. As suggested, we have listed all the MS identified potential acetylation sites of Stat6 in the revised **Figure 1g**, including K73, K374, K383 and K636. Among these sites, K383 locate in the DNA-binding domain (DBD) of Stat6 protein. In addition, we have presented the MS images for all the identified potential acetylation sites in the revised **Supplementary Figure 1**.

- *Fig 2C: The acK383 Antibody is apparently not fully specific, since substantial signal is still present in the mutant – this issue should be discussed.*

Response

We agree with the reviewer that the anti-acK383 Stat6 polyclonal antibody is not fully specific, and discussed this point in the revised Result section as follows:

Indeed, this antibody recognized CBP-induced acetylation in Stat6 but not in K383R mutant Stat6, although there is a basal background signal due to unspecific reaction (revised **Figure 2c**).

- *Fig 2D-E: Detailed description of FACS analysis is necessary. Appropriate controls should be shown. Cell type is not clear. Was the experiment performed in macrophages from tumor? NT should be disabbreviated.*

Response

Following the reviewer's suggestion, we have added the appropriate isotype controls in this experiment and showed the data in the revised **Figure 2d-e**. The cells used in this experiment are the mouse peritoneal macrophages that isolated from naïve C57BL/6 mice, which has been indicated in the revised legend for **Figure 2d-e**. We have also revised the "NT" label with "DMSO", since this group of the cells were treated with vehicle DMSO. In addition, we have provided the detailed description of FACS analysis for Stat6 acetylation in the revised Method section as follows:

For Stat6 K383 acetylation analysis, the collected macrophages were fixed with Fixation/Permeabilization Buffer (Invitrogen) on ice for 20 min, washed with Permeabilization Buffer (Invitrogen) and spin down at 3000 rpm for 5 min at 4 °C. Cells were then resuspended and incubated on ice with anti-Ac-Stat6 (K383) or isotype IgG control antibody for 1 h, followed by washing with Permeabilization Buffer and spin down at 3000 rpm for 5 min at 4 °C. Cell precipitate was resuspended with PE-conjugated anti-rabbit secondary antibody (Invitrogen) in Permeabilization Buffer and incubated on ice for 1 hour. After washing with Permeabilization Buffer and spin down at 3000 rpm for 5 min at 4 °C, the cell was resuspended

with Permeabilization Buffer and analyzed using a Beckman Gallios or BD Aria2 flow cytometer. The flow cytometry data were analyzed using FlowJo software.

- *Fig 2F: Why the K383R mutant shows big increase of activity without CBP overexpression? (no acetylation should be expected). Control for protein expression levels of different mutant constructs should be shown.*

Response

We apologize for not indicating clearly in the original figure 2f. Actually, we have overexpressed the CBP protein for all the groups in this experiment, and presented all the protein expression immunoblot panels, including CBP, different Stat6 mutant and loading control, below the bar graph.

- *Fig 2H: pSTAT6 and STAT6 appear sometimes as a doublet here and several other blots. Preferably only one band should be presented to avoid confusion.*

Response

As suggested, we have presented only one band for P-Stat6 and Stat6 in the revised figures.

- *Fig 2I: A western blot verifying equal levels of STAT6 expression should be shown as well as the effect of the mutant without IL-4 stimulation.*

Response

Following the reviewer's suggestion, we have presented the western blot panels verifying equal levels of wild-type (WT) and K383R mutant STAT6 expression between groups in the revised **Figure 2j**.

- *Lane 163-166: It is unclear how these analyses were conducted – reveal the details and the source of data.*

Response

We apologize for the confusion in the initial submission. The text between line 163 to 166 described the data in **Figure 3a**, which explained how Trim24 was selected for further mechanistic analysis of CBP-mediated Stat6 acetylation. Because Stat6 acetylation is induced in macrophages upon IL-4 stimulation, we hypothesized that IL-4-controlled genes may be implicated in CBP-mediated Stat6 acetylation. Therefore, we performed the RNA-sequencing to compare the differential expression genes between non-treated and IL-4-treated macrophages, and identified there were total 1613 genes that up/down-regulated in macrophages upon IL-4 stimulation. In addition, the CBP-associated proteins may be the regulator of CBP-mediated Stat6 acetylation. Thus, we performed the mass spectrum analysis to identify potential CBP-interacting protein after immunoprecipitation of CBP. Among the 1613 genes that identified by RAN-seq, we found that there were 63 genes that were associated with CBP as well. We hypothesized that these 63 genes might be the candidate to regulate CBP-mediated Stat6 acetylation. Among these genes, we selected the most abundantly expressed genes that also have reported function in macrophages previously. Finally, Trim24, a bromodomain-containing protein that can recognize acetylated lysine residues, was chosen for further analysis. We have modified the sentences in the Results section of the revised manuscript as suggested.

In addition, the source data for RNA-seq is uploaded in Gene Expression Omnibus database with the accession code of GSE116588, and the source data for RNA-seq and MS analysis of CBP-associated proteins were included as Supplementary data in the revised manuscript.

- *Fig 3B: For me, Trim24 is partly nuclear independently of IL-4 stimulation. Nuclear fractionation western blot is necessary to make quantitative claims on Trim24 nuclear translocation upon IL-4 stimulation.*

Response

Per reviewer comment, we examined the Trim24 protein levels in cytosol and nucleus of macrophages with or without IL-4 stimulation by western blot. The results revealed that IL-4 does not affect the subcellular localization of Trim24, which is mainly localized in the nucleus of macrophages with or without IL-4 stimulation. The data is included in the revised **Supplementary Figure 2a**.

- *Fig 3C: In this experiment, acLys and Flag images appear to be from different blots. The increase in acLys upon Trim24 is not obvious. For this experiment, please provide the full-length scans of all western blots (at least three biological replicates) with indication of molecular weight marker positions.*

Response

Following the reviewer’s comment, we have provided the original source data for this experiment, which suggested that Trim24 indeed promote CBP-mediated Stat6 acetylation on a dose-dependent manner. The data of three biological replicates were as follows:

- *Fig 3D: In this experiment (as well as in some others) Trim24 seems to affect total levels of CBP. The effect of Trim24 on total CBP levels should be addressed.*

Response

We thank the reviewer for this comment. CBP is a very large protein with a molecular weight of about 300 kD, so sometimes the detection of CBP protein expression by immunoblot may have some variations. In our present study, we found that *Trim24* deficiency or knockdown does not affect the endogenous CBP protein levels in both mouse primary macrophages and iBMDM (the input CBP immunoblot in revised **Figure 5a** and **Figure 4c**). In addition, neither *Trim24* overexpression with different dose nor *Trim24* knockdown in 293T cells affect the protein expression levels of transfected HA-CBP (revised **Figure 3c, 3h, 3i**). These data collectively suggested that *Trim24* does not affect the protein levels of endogenous and overexpressed CBP. Therefore, we repeated this experiment for Figure 3d, and presented that

data with equal HA-CBP expression with or without Trim24 overexpression in input sample (revised **Figure 3d**).

- *Fig 3E: Here the promotion of the STAT6-CBP complex by Trim24 is not obvious since: a) CBP levels appear to be also increased by Trim24 OE and b) amount of Flag-STAT6 in the IP seems to be different with and without Trim24 overexpression. For this experiment, please provide the full-length scans of all western blots (at least three biological replicates) with indication of molecular weight marker positions.*

Response

We thank the reviewer for this comment. After carefully repeated this experiment, we concluded that Trim24 overexpression does not affect HA-CBP protein levels. In the revised **Figure 3e**, the input HA-CBP protein levels are equal with or without Trim24 overexpression. However, CBP overexpression seemed to upregulate Flag-Stat6 protein expression in 293T cells due to artificial effect of too much CBP protein, because CBP knockdown dose not affect endogenous Stat6 protein levels in mouse macrophages. So this is the reason why there are more input and immunoprecipitated Flag-Stat6 in the last two lanes of CBP-overexpressed groups. Nevertheless, Trim24 overexpression does not affect Flag-Stat6 expression, but promoted the Stat6-CBP association when the protein levels of input CBP and immunoprecipitated Stat6 are equal in the last two lanes of this figure (revised **Figure 3e**). More importantly, we also presented the data in revised **Figure 3f** that *Trim24* knockdown inhibited the association of CBP with Stat6, which further supported our conclusion that Trim24 indeed promoted Stat6-CBP association.

Together with this submission, we provided the full-length scans of source data with indication of molecular weight marker. In addition, the representative three biological replicates for this experiment were as follows:

- *Fig 3F: Please show Trim24 levels after knockdown in input samples.*

Response

As suggested, Trim24 immunoblot panel was included in the revised **Figure 3f** as suggested.

- *Fig 3G: Please use western blot analysis in this experiment. FACS analysis may be used only if the sample amount is unsuitable for western analysis, which is not the case here.*

Response

We have replaced the FACS data with western blot analysis showing the decrease of Stat6 K383 acetylation in *Trim24* knockdown cells as suggested.

- *Fig 3H-I: In these both cases it is unclear how Trim24 achieves effects on STAT6 without overexpressed CBP. Please show the effects on STAT6 acetylation for both overexpression and knockdown of Trim24.*

Response

We apologize for the inaccurate presentation of the figures in the initial submission. Actually, we have overexpressed CBP to examine Stat6 acetylation and related luciferase activity in 293T cells with Trim24 overexpression or knockdown. The immunoblot panels for Stat6 acetylation, CBP and Trim24 have been included in the revised **Figure 3h-i**.

- *Fig 4C: The effect of IL-4 on CBP Ubiquitination is difficult to judge due to the difference in CBP levels in the IP. For this experiment, please provide the full-length scans of all western blots (at least three biological replicates) with indication of molecular weight marker positions. Please also provide the details of the Anti-Ub western blot experiments in the experimental procedures section.*

Response

Following the reviewer's suggestion, we have provided the original source data with indication of molecular weight marker positions for this experiment, which suggested that *Trim24* knockdown indeed inhibited IL-4-induced CBP ubiquitination. In addition, the detailed protocol for the anti-Ub western blot experiments was included in the Method section in the revised manuscript as follows:

For *in vivo* ubiquitination assay, macrophages that left unstimulated or stimulated with IL-4, or 293T cells that transfected with the desired plasmids were lysed with cell lysis buffer containing protease inhibitor and N-ethylmaleimide. After saving some cell extracts as input analysis, the remaining cell extracts were then boiled at 100 °C for 5 min in the presence of 1% SDS and then diluted with RIPA buffer till the SDS concentration was 0.1%. The diluted lysates were pre-cleaned with Protein-A/G-coupled agarose beads, then incubated with specific immunoprecipitation antibody on the shaker at 4 °C overnight. The next day, the immunoprecipitated proteins were collected by incubation with Protein-A/G-coupled agarose beads on the shaker at 4 °C for 2 h, washed with RIPA buffer containing protease inhibitors, PMSF and N-ethylmaleimide, boiled at 100 °C for 5 min, and then loaded to run SDS-PAGE. The immunoprecipitates were immunoblotted with anti-ubiquitin or indicated antibodies.

The data of three biological replicates were as follows:

- *Fig 4F: Please present the results of the mass-spectrometric analysis in a way clear and*

understandable to a non-specialist. More details and the description are needed.

Response

Following the reviewer's suggestion, we modified this figure and presented the result of the mass-spectrometric analysis in revised **Figure 4f**, showing K119 is the ubiquitination site of CBP in a way clearer and more understandable. In addition, we have incorporated more detailed description of MS analysis of CBP ubiquitination site in the revised legend for this figure, and provided the original mass spectrum data in the Source data file.

- *Fig 4I: It looks like the amount of Flag-STAT6 in the input is not equal, which makes questionable the effect of Trim24 overexpression on STAT6 acetylation in this experiment.*

Response

We thank the reviewer for this comment. In this study, we found that CBP overexpression in 293T cells could enhance Flag-Stat6 expression. However, CBP knockdown does not affect the endogenous Stat6 protein level in macrophages (revised **Figure 1c**), suggesting the enhanced Flag-Stat6 expression may be due to the artificial effect of CBP overexpression in 293T cells. Therefore, we removed the 293T cell group that without CBP overexpression, and just compared the Stat6 acetylation levels in all groups of 293T cells with CBP overexpression, which showed comparable protein levels of Flag-Stat6 both in input and immunoprecipitated samples (revised **Figure 4i**). Under this condition, Trim24 overexpression indeed promoted WT CBP-induced but could not enhance K119R mutant CBP-induced Stat6 acetylation (revised **Figure 4i**). These results suggested that CBP-mediated Stat6 acetylation is dependent on Trim24-induced CBP ubiquitination at K119.

- *Fig 4J: Why is HA-Trim24 signal present in the second lane where no HA-Trim24 should be?*

Response

We thank the reviewer for this comment. Indeed, Trim24 should not be presented in the second lane since there is no Trim24 expression vector transduction for this group. In this experiment, both CBP and Trim24 vector contain HA tag, so the appeared band of second lane that has similar molecular weight as Trim24 may be come from nonspecific reaction or the short isoform of HA-CBP when immunoblotting with anti-HA antibody. To exclude these possibilities, we replaced HA tagged Trim24 with Myc tagged Trim24 to repeat this experiment, and there is no nonspecific or short isoform of HA-CBP after immunoblotting with anti-Myc antibody in the lane that without Trim24 overexpression (revised **Figure 4j**). As we responded in the above-mentioned comment that CBP overexpression would artificially affect Flag-Stat6 expression in 293T cells, we removed the first lane that without CBP overexpression (revised **Figure 4j**).

- *Fig 5A: CBP blots both in input and in the IP samples are of very poor quality, please provide better quality blots.*

Response

We thank the reviewer for this comment. CBP is a very large protein with a molecular weight of nearly 300 kD, so technically it is very difficult to examine the endogenous ubiquitination of CBP especially in primary cells, as reflected by very few paper showing the

result of endogenous CBP ubiquitination. In **Figure 5a**, although the CBP blots are of poor quality, there are much more endogenous CBP ubiquitination detected in IL-4-treated WT macrophages as compared with that of *Trim24*-deficient macrophages upon IL-4 stimulation, even there is higher level of immunoprecipitated CBP in the lane of IL-4-treated *Trim24*-deficient macrophages. We also quantified the endogenous ubiquitinated CBP by using ImageJ software, showing the values of IL-4-treated WT vs. *Trim24*-deficient macrophages is 1.76 vs. 0.33 (revised **Figure 5a**). In addition, we also presented the data in revised **Figure 4c**, showing *Trim24* knockdown dramatically suppressed IL-4 induced endogenous CBP ubiquitination in iBMDM.

- *Fig 5B: The fact that IL-4 still enhances STAT6 acetylation in Trim24 knockout macrophages should be discussed.*

Response

We thank the reviewer for this comment, and discussed this point in the Results section of the revised manuscript as suggested in the following:

Actually, we also observed that Stat6 is slightly acetylated in *Trim24*-deficient macrophages upon IL-4 stimulation (revised **Figure 5b**), implying other acetyltransferases like p300 may function redundantly to mediate Stat6 acetylation in the absence of Trim24-mediated CBP activation.

- *Fig 5F-G: The usage of TSA/NAM in these experiments should be avoided.*

Response

We have deleted the data by using TSA/NAM as suggested.

- *Fig 5H-I: Please use western blot analysis in this experiment. FACS analysis may be used only if the sample amount is unsuitable for western analysis, which is not the case here.*

Response

We thank the reviewer for this comment. We didn't replace this FACS data of Stat6 acetylation examination with western blot analysis due to the following reason.

To examining the endogenous Stat6 acetylation by using western blot experiment, we have to prepare at least 200 million macrophages for each group, because we need to immunoprecipitate the endogenous Stat6, and then immunoblot with anti-acK383 Stat6 antibody. However, only 0.5 million macrophages can be collected from 15 ml human peripheral blood. Therefore, it is difficult to prepare enough cells to examine the endogenous Stat6 K383 acetylation by using western blot analysis in human PBMC-derived macrophages. Nevertheless, we have provided the western blot data in both *Trim24*-knockdown iBMDM (revised **Figure 3g**) and *Trim24*-deficient mouse primary macrophages revised (**Figure 5b**), showing that *Trim24* deficiency indeed suppressed endogenous Stat6 K383 acetylation. In addition, this FACS data also in concert with the western blot data that collected from *Trim24*-knockdown iBMDM and *Trim24*-deficient mouse primary macrophages. Therefore, we hope the reviewer would consider this situation, thank you!

- *Fig 5J. Please expand the panel of IL-4-dependent genes regulated by Trim24 in human macrophages.*

Response

Following the reviewer's comment, we examined the expression of more IL-4-dependent M2 genes, including *TGM2*, *CCL17*, *IRF4*, in human macrophages, and found that TRIM24 knockdown also promoted these M2 gene's expression in human macrophages. The related data were presented in revised **Figure 5m**.

- *Fig 6J: Ccl24 is reduced in the macrophages, not in the tumor. How this can explain reduction of macrophage infiltration in the tumor?*

Response

We thank the reviewer for this comment. Our data suggested that *Trim24* deficiency promoted the expression of M2 genes, including macrophage-recruiting chemokine *Ccl24*, in both primary cultured macrophages (revised **Figure 5j**) and tumor-infiltrated macrophages (revised **Figure 6j**). As a consequence, the increased CCL24 may promote the macrophage infiltration in the tumor of *Trim24*^{M-/-} mice (revised **Figure 6k**).

- *Fig 6K: Bar chart Y-axis description is the same for two graphs – should lower graph depict the levels of CD8+ cells?*

Response

We apologize for this error, and corrected with “CD8⁺ cells” in the Y-axis description of the bar chart in Figure 6l.

- *Fig 7D: Why THP-1 and not PBMCs are used in this case?*

Response

Following the reviewer's suggestion, we performed the experiment to examine TRIM24 expression in PBMC-derived macrophages upon IL-4 stimulation, which showed similar tendency as that in THP-1 cells. The data that collected from PBMC-derived macrophages has been included as revised **Figure 7c, 7d** in the revised manuscript.

REVIEWERS' COMMENTS:

Reviewer #1 (Remarks to the Author):

In the revised manuscript the authors have addressed my concerns by providing global gene expression data from Trim24 deficient cells. They also show increased STAT6 binding in these cells. Finally they used normal and tumor associated macrophages to show altered M2 marker genes. In this latter part the number of samples are rather low (6-6), but the trend appears to be clear.

Reviewer #2 (Remarks to the Author):

The authors addressed all the concerns raised in the original review. There are only two minor corrections to the manuscript text which need to be made.

Lane 169: Venn graph should be changed to Venn diagram

Figure 5g STAT6 binding to the Ym1 promoter is indicated on the figure, but Irf4 is in the figure legend

Point-by-point response letter

Reviewer #1 (Remarks to the Author):

- *In the revised manuscript the authors have addressed my concerns by providing global gene expression data from Trim24 deficient cells. They also show increased STAT6 binding in these cells. Finally they used normal and tumor associated macrophages to show altered M2 marker genes. In this latter part the number of samples are rather low (6-6), but the trend appears to be clear.*

Response

We thank the reviewer for the positive comments and the insightful suggestions. Although the QPCR validation of M2 genes number is low, the RNA-seq data indicated that Trim24 deficiency indeed promoted M2 signature gene expression, which were enriched in *Trim24*-knockout cells through gene set enrichment analysis (GSEA) as showed in revised Figure 6b.

Reviewer #2 (Remarks to the Author):

- *The authors addressed all the concerns raised in the original review. There are only two minor corrections to the manuscript text which need to be made.*

Response

We thank the reviewer for the positive comments and have made the corrections as suggested by the reviewer.

- *Lane 169: Vein graph should be changed to Venn diagram*

Response

Following the reviewer's suggestion, we have changed the description of "Vein graph" to "Venn diagram" throughout the manuscript.

- *Figure 5g STAT6 binding to the Ym1 promoter is indicated on the figure, but Irf4 is in the figure legend*

Response

We apologize for this error, and have changed "Irf4" to "Ym1" in the figure legend for Figure 5g, which is consistent with the information in the related figure. The corrected figure legend was highlighted in red.